# Aligning Tree-Search Policies with Fixed Token Budgets in Test-Time Scaling of LLMs

**Sora Miyamoto** [1]  **Daisuke Oba** [1]  **Naoaki Okazaki** [1 2 3]

## Abstract

Tree-search decoding is an effective form of test-time scaling for large language models (LLMs), but real-world deployment often imposes a fixed per-query token budget that varies across settings. Existing tree-search policies are largely budget-agnostic, treating the budget merely as a termination condition, thereby risking late-stage over-branching or premature termination. We propose Budget-Guided MCTS (BG-MCTS), a tree-search decoding algorithm that aligns its search policy with the remaining token budget: it starts with broad exploration, then prioritizes refinement and answer completion as the remaining budget decreases while reducing late-stage branching from shallow nodes. BG-MCTS consistently outperforms budget-agnostic tree-search baselines across inference budgets on mathematical reasoning benchmarks and an additional physics reasoning benchmark with open-weight LLMs.

🐙 github.com/Sora-Miyamoto/bg-mcts

## 1. Introduction

Response quality for Large Language Models (LLMs) can be improved either by optimizing model parameters (Vaswani et al., 2017; Brown et al., 2020; Ouyang et al., 2022) or improving inference while keeping the parameters fixed. The latter is known as *Test-Time Scaling* (Zhang et al., 2025).

Test-time scaling methods are often grouped into *parallel* sampling-and-aggregation (Wang et al., 2023; Brown et al., 2024; Snell et al., 2024; Lightman et al., 2024), *sequential*

[1]Department of Computer Science, Institute of Science Tokyo, Japan [2]AIRC, National Institute of Advanced Industrial Science and Technology (AIST), Japan [3]LLMC, National Institute of Informatics (NII), Japan. Correspondence to: Sora Miyamoto <sora.miyamoto@nlp.comp.isct.ac.jp>.

*Proceedings of the 43rd International Conference on Machine Learning*, Seoul, South Korea. PMLR 306, 2026. Copyright 2026 by the author(s).

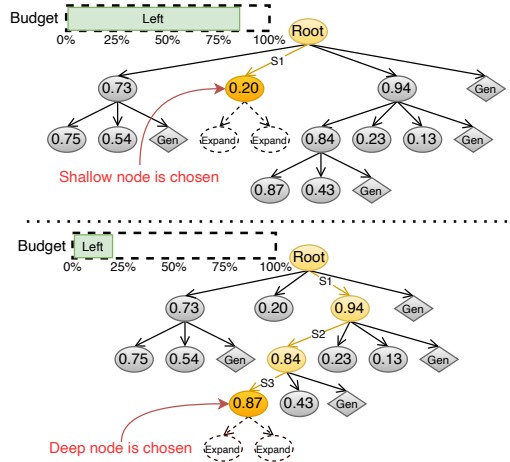

*Figure 1.* **Conceptual diagram of node selection in BG-MCTS**. With a large remaining budget, the policy favors shallow nodes for broad exploration (**Top**). As the remaining budget decreases, it favors deeper nodes to refine promising candidates (**Bottom**).

refinement conditioned on the previous sampling (Madaan et al., 2023; Yao et al., 2023b; Muennighoff et al., 2025), and *hybrid* approaches that combine both (Yao et al., 2023a; Wang et al., 2024b; Besta et al., 2024; Tian et al., 2024; Zhou et al., 2024; Xie et al., 2024; Zhang et al., 2024; Wan et al., 2024; Chen et al., 2024; Chang et al., 2025; Inoue et al., 2025; Zheng et al., 2025). Hybrid methods often use *tree-search decoding*, branching into multiple continuations and expanding the most promising ones. Increasing test-time compute budget allows the search to explore more alternatives, often yielding better final responses.

Still, deployment imposes a fixed per-query inference budget, which can vary widely across products and settings. The objective is therefore *to make the most of the available budget to maximize answer quality*.

Yet most existing tree-search decoding policies are largely budget-agnostic: they either use fixed search hyperparameters, such as the number of iterations, branching factor, and search width/depth (Yao et al., 2023a; Tian et al., 2024; Inoue et al., 2025), or treat the budget merely as a stopping condition. This budget mismatch leads to two common failure modes: the search may over-branch late and exhaust

the budget before refinement or verification, or it may stop prematurely and leave part of the budget unused. Early-stopping variants (Wang et al., 2024a) focus on deciding when to stop in order to save tokens, but do not define a budget-conditioned policy for shifting search from branching to refinement as the remaining budget decreases.

We propose **Budget-Guided MCTS (BG-MCTS)**, a tree-search decoding algorithm that *aligns* its search policy with a fixed *token budget*—the total number of output tokens generated across the entire search (Fig. 1). Building on Monte Carlo Tree Search (MCTS) (Silver et al., 2017), BG-MCTS makes *budget-dependent* decisions for node selection and expansion by conditioning them on the remaining budget. As a result, BG-MCTS begins with broad exploration to avoid premature commitment, and gradually shifts toward refining and completing the most promising candidates as the budget decreases, while suppressing unnecessary new branches. This budget-aligned wide-to-deep behavior enables the search to first *think broadly* and then *finish strong*, turning the same token budget into more reliable solutions.

We evaluate BG-MCTS on two mathematical reasoning benchmarks, MATH500 (Lightman et al., 2024) and AIME24/25 (Maxwell-Jia, 2024; OpenCompass, 2025). As an additional test beyond mathematical reasoning, we evaluate BG-MCTS on one physics reasoning benchmark, UG-Physics (Xu et al., 2025). We use widely available open-weight LLMs, including Llama-3.1-8B-Instruct (Grattafiori et al., 2024), Qwen2.5-7B-Instruct (Qwen et al., 2025), and Qwen3-32B (Team, 2025), with UGPhysics evaluated on the two Qwen models. Across per-instance token budgets $B \in \{10\text{k}, 20\text{k}, 30\text{k}\}$, BG-MCTS consistently outperforms budget-agnostic tree-search baselines.

**Outline.** The remainder is organized as follows: § 2 provides preliminaries, § 3 introduces BG-MCTS, § 4 reports experimental results, § 5 presents our analysis, § 6 discusses implications and limitations, and §7 discusses related work.

## 2. Preliminaries

**Tree-search decoding.** We study *tree-search decoding* for LLMs, which builds a search tree over partial generations. Each node represents a partial output (prefix) and optionally its intermediate reasoning state; expanding a node generates one or more continuations. We denote a parent node by $p$ and a child by $s \in \mathcal{S}(p)$, where $\mathcal{S}(p)$ is the current set of children of $p$. The search iterates selection, expansion, and evaluation, and returns the best completed answer found.

**Token budget.** We focus on the fixed-budget setting where each problem instance is run under a fixed *token budget* $B$. Let $C_{\text{used}}$ denote the cumulative number of output tokens generated by the LLM across all expansions. The search must satisfy $C_{\text{used}} \leq B$ and terminates when the budget is

exhausted. A budget-aware policy should therefore account for the remaining budget when deciding whether to branch further or deepen existing candidates.

**Node evaluation and statistics.** When a node $x$ is expanded, it is assigned a scalar evaluation $Q(x) \in \mathbb{R}$, e.g., from a verifier/reward model. Let $\mathcal{T}(x)$ denote the set of expanded nodes in the subtree rooted at $x$. For each node $x$, we maintain an accumulated value and a subtree size

$$W(x) = \sum_{y \in \mathcal{T}(x)} Q(y), \qquad m_x = |\mathcal{T}(x)|,$$

Here, $m_x$ is defined as subtree size rather than visit count for simplicity.[1] We keep $W(\cdot)$ explicit because our method later modifies the value accumulation rule.

**Child priors.** Tree-search decoding often uses a prior distribution $P(s \mid p)$ over children to guide exploration, for example, from a learned policy network (Silver et al., 2017). When such priors are unavailable, a practical alternative is to construct them from available scores for candidate children (Inoue et al., 2025). In our experiments, following Inoue et al. (2025), we instantiate

$$P(s \mid p) = \text{softmax}(\{ Q(s') \mid s' \in \mathcal{S}(p) \})_s .$$

Our method is compatible with other choices of $P$.

**MCTS with PUCT.** Standard MCTS repeats *selection*, *expansion*, *evaluation*, and *backpropagation* (Silver et al., 2017; Inoue et al., 2025; Chen et al., 2024). During selection, starting from the root, the search repeatedly chooses the child $s \in \mathcal{S}(p)$ that maximizes:

$$\text{PUCT}(p, s) = \underbrace{\frac{W(s)}{m_s}}_{\text{Exploitation}} + \underbrace{c \, P(s \mid p) \sqrt{\frac{\ln(m_p)}{m_s}}}_{\text{Exploration}}, \quad (1)$$

where $c > 0$ controls the exploration intensity. After expanding and evaluating new nodes, backpropagation updates $W(\cdot)$ and $m(\cdot)$ along the selected path.

**Budget-agnostic vs. Budget-aware policies.** Eq. 1 is budget-agnostic: the selection score depends only on tree statistics and priors, while the token budget $B$ is used merely as a stopping condition when $C_{\text{used}}$ reaches $B$. We instead seek budget-aware policies that condition search decisions on the remaining budget $B - C_{\text{used}}$ to allocate a fixed budget more effectively.

## 3. Budget-Guided MCTS

We propose **Budget-Guided MCTS (BG-MCTS)**, a tree-search decoding algorithm for fixed-budget test-time scaling (Figure 1). BG-MCTS *aligns* MCTS decisions with a pre-specified *token budget* by conditioning the search policy

---

[1] In some implementations, $m_x$ is defined as a visit count.

on the *remaining budget*. Concretely, starting from standard PUCT-style MCTS (Sec. 2), BG-MCTS introduces two budget-conditioned control mechanisms that adapt the search *behavior*: (i) it adjusts the **selection dynamics** via a budget-conditioned PUCT score, and (ii) it **regulates tree widening** through a budget-guided mechanism that decides when to introduce new children. Together, these mechanisms induce a budget-aligned wide-to-deep schedule: the search stays exploratory early and increasingly concentrates on refining and completing promising candidates as the budget runs down.

## 3.1. Budget and Cost Tracking

Let $B$ be the total token budget for a problem instance. Let $C_{\text{used}}$ be the cumulative number of output tokens generated by the LLM across the entire search so far (Sec. 2). We use the *budget sufficiency ratio* $\rho \in [0, 1]$,

$$\rho \;=\; 1 - \frac{C_{\text{used}}}{B}, \tag{2}$$

as the conditioning variable of the search policy. Intuitively, $\rho \simeq 1$ corresponds to the early stage (budget ample) and $\rho \simeq 0$ corresponds to the late stage (budget nearly exhausted).

## 3.2. Budget-Guided Selection

In standard PUCT (Eq. 1), the selection score is independent of $B$ and $\rho$, and the budget typically affects the search only through termination. BG-MCTS makes selection budget-conditioned by *annealing* the exploration bonus with $\rho$ while simultaneously adding a late-stage completion bias to the value term; as $\rho$ decreases, the influence of the exploration term diminishes, biasing selection toward nodes with higher mean values.

**BG-PUCT score.** Given the budget sufficiency ratio $\rho$, BG-MCTS selects the child for a parent $p$ and a standard child $s \in \mathcal{S}_{\text{std}}(p)$ that maximizes,

$$\text{BG-PUCT}(p, s, \rho) = \underbrace{\frac{\tilde{W}(s, \rho)}{m_s}}_{\text{Exploitation}} + \underbrace{\rho\, c\, P(s \,|\, p)\sqrt{\frac{\ln(m_p)}{m_s}}}_{\text{Exploration}}, \tag{3}$$

where $c > 0$ is the exploration weight, $P(s \,|\, p)$ is the child prior, and $m_s$ is the subtree size (Sec. 2). The key differences from Eq. 1 are: the accumulated value also depends on the budget sufficiency ratio ($W(s) \to \tilde{W}(s, \rho)$); and the multiplicative factor $\rho$ is incorporated in the exploration term (exploration is gradually suppressed as $\rho$ decreases).

**Budget-conditioned depth-biased value correction.** We define the corrected accumulated value $\tilde{W}(s, \rho)$ as

$$\tilde{W}(s, \rho) = \sum_{x \in \mathcal{T}(s)} \tilde{Q}(x, \rho), \tag{4}$$

$$\tilde{Q}(x, \rho) = Q(x) + \underbrace{\kappa(1 - \rho)\frac{d(x)}{\hat{d}_{\text{ans}}}}_{\text{completion bias}}, \tag{5}$$

where $\mathcal{T}(s)$ denotes the expanded subtree rooted at $s$, $\kappa \geq 0$ is a constant, and $d(x)$ denotes the depth of node $x$. For nodes that already contain a completed answer, we set this term to zero, since their quality is directly determined by the final-answer evaluation and should not be further amplified by the exploitation term. The term $\hat{d}_{\text{ans}}$ denotes an estimate of the depth at which an answer is typically completed; in practice, we use the running average depth of nodes that contain a completed answer (or the current maximum expanded depth if no answer has been found yet). The correction is scaled by $(1-\rho)$, so it is negligible early and becomes more influential as the budget runs out.

**Implication.** When $\rho \simeq 1$, BG-PUCT is close to standard PUCT: it does not artificially boost exploration, but it also does not prematurely damp it. As $\rho$ decreases, the exploration bonus is annealed and the completion shaping in Eq. 5 becomes stronger, shifting selection away from opening new alternatives and toward deepening/refining a few promising branches. Combined with the budget-guided widening mechanism (Sec. 3.3), this yields a wide-to-deep search schedule over the course of the budget.

## 3.3. Budget-Guided Tree Widening

Selection alone cannot prevent a common fixed-budget failure mode: introducing new alternatives too late to meaningfully refine them. Many tree-search decoders therefore use *dynamic widening* (Inoue et al., 2025), allowing the search to generate additional children from intermediate nodes as the search proceeds. While this flexibility helps adapt where the tree expands, it can be wasteful near the end of a fixed-budget search, since newly introduced branches may not receive enough remaining tokens to become useful. Instead, BG-MCTS makes widening itself budget-aware.

**Virtual generative child.** For each non-terminal node $p$, let $\mathcal{S}_{\text{std}}(p)$ denote its current set of standard, actual children. BG-MCTS augments this set with a virtual generative child $s_{\text{gen}}(p)$ and defines the selectable set

$$\mathcal{S}(p) = \mathcal{S}_{\text{std}}(p) \cup \{s_{\text{gen}}(p)\}.$$

The virtual child represents the action of generating a new child from $p$. If $s_{\text{gen}}(p)$ is selected, BG-MCTS does not expand the virtual node itself; instead, it samples an additional standard child from $p$ and adds it to $\mathcal{S}_{\text{std}}(p)$. Thus,

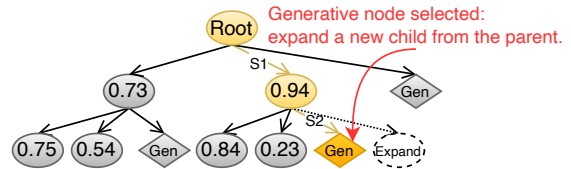

*Figure 2.* **Tree widening in BG-MCTS.** When the virtual generative child (diamond) is selected, it generates a new standard child from its parent instead of expanding the virtual node itself.

widening becomes a first-class option that competes with selecting existing children (Figure 2).

**Generative score.** To choose between deepening existing children and widening at $p$, BG-MCTS assigns the generative option the following score

$$E_{\text{gen}}(p,\rho) = \underbrace{\mu(p)}_{\text{value level}} + \lambda\,\rho\,\underbrace{\sigma^2(p)}_{\text{uncertainty}}, \qquad (6)$$

where $\mu(p)$ and $\sigma^2(p)$ denote the mean and variance of $Q(s)$ over $s \in \mathcal{S}_{\text{std}}(p)$, respectively, and $\lambda \geq 0$ controls the exploration–exploitation trade-off.

If $\mathcal{S}_{\text{std}}(p) = \emptyset$, the generative option is selected by default. The mean term favors widening from promising nodes, while the variance term promotes widening when existing children disagree. Multiplication by $\rho$ makes this incentive strong early in the search and gradually suppresses widening as the remaining budget decreases. This discourages late-stage branching from shallow nodes and helps convert earlier exploration into refinement and completed solutions.

### 3.4. Unified Selection with Widening Trigger

BG-MCTS follows the standard MCTS loop (Sec. 2), but makes both deepening and widening decisions budget-aware. At each internal node $p$, *selection* is performed over the augmented child set $\mathcal{S}(p)$:

$$s^\star \in \arg\max_{s \in \mathcal{S}(p)} \text{Score}(p, s, \rho),$$

where

$$\text{Score}(p, s, \rho) = \begin{cases} \text{BG-PUCT}(p, s, \rho), & s \in \mathcal{S}_{\text{std}}(p), \\ E_{\text{gen}}(p, \rho), & s = s_{\text{gen}}(p). \end{cases} \quad (7)$$

If $s^\star$ is a standard child, the search descends to that child; if $s^\star = s_{\text{gen}}(p)$, BG-MCTS widens $p$ by generating a new standard child. Because both BG-PUCT and the widening trigger are conditioned on $\rho$, exploration and widening are gradually annealed as the remaining budget decreases, shifting the search from broad exploration toward refinement.

### 3.5. Algorithm

Algorithm 1 summarizes the overall BG-MCTS procedure. We briefly explain the subroutines for completeness.

---

**Algorithm 1 BG-MCTS**

---

**Notation:** $\Delta\mathcal{S}(p)$ is newly expanded children from $p$ in EXPAND; $\Delta C$ is the number of output tokens generated to produce $\Delta\mathcal{S}(p)$.

**Require:** root node $p_0$, token budget $B$, leaf expansion width $k$
1: Initialize tree $T$ with $p_0$; $C_{\text{used}} \leftarrow 0$
2: **while** $C_{\text{used}} < B$ **do**
3:  $\rho \leftarrow 1 - C_{\text{used}}/B$
4:  $p \leftarrow \text{SELECT}(T, \rho)$
5:  $(\Delta\mathcal{S}(p), \Delta C) \leftarrow \text{EXPAND}(T, p, k)$
6:  $Q(s) \leftarrow \text{EVALUATE}(s)$ for all $s \in \Delta\mathcal{S}(p)$
7:  $\text{BACKPROP}(T, \Delta\mathcal{S}(p))$
8:  $C_{\text{used}} \leftarrow C_{\text{used}} + \Delta C$
9: **end while**
10: **return** best completed answer found in $T$

---

**SELECT**$(T, \rho)$. Starting from the root, repeatedly apply Eq. 7. If the maximizer is a standard child, descend to that child. Return either the reached leaf for ordinary expansion, or the first non-terminal node $p$ where $s^\star = s_{\text{gen}}(p)$.

**EXPAND**$(T, p, k)$. If $p$ is a leaf, generate $k$ children. If $p$ is selected through its generative child, generate one additional standard child from $p$ to widen the tree. Return the newly added children $\Delta\mathcal{S}(p)$ and their generated-token cost $\Delta C$.

**EVALUATE**$(s)$. Assign a scalar value $Q(s)$ to each newly added child $s$, using a verifier, reward model, or heuristic. If $s$ contains a completed answer, update $\hat{d}_{\text{ans}}$ in Eq. 5.

**BACKPROP**$(T, \Delta\mathcal{S}(p))$. For each $s \in \Delta\mathcal{S}(p)$, update subtree statistics along the path from $s$ to the root: $m_a \leftarrow m_a + 1$ and $W(a) \leftarrow W(a) + Q(s)$ for each ancestor $a$.

## 4. Experiments

### 4.1. Experimental Setup

**Fixed-budget protocol.** We measure inference cost by the total number of *output tokens* generated across the entire search, denoted $C_{\text{used}}$. Each instance is run under a fixed token budget $B \in \{10\text{k}, 20\text{k}, 30\text{k}\}$, and search terminates once $C_{\text{used}} \geq B$. Among all nodes containing a valid final answer, we return the one with the highest score $Q(\cdot)$.[2] Correctness is evaluated using LightEval (Habib et al., 2023).

**Node construction.** We construct nodes using two generation units: (i) **Full generation**, where the model generates until a stop token or a maximum context length, or (ii) **Sequential generation**, where generation is truncated at predefined step boundaries; we use "\nStep" to mark step boundaries. To continue search from nodes that already contain an answer, we append the prompt "But wait, let me think about the problem again." and resume (Muennighoff et al., 2025).

---

[2]We extract candidate answers with dataset-specific regular expressions; details are provided in Appendix A.

**Node evaluation.** Each newly expanded node $x$ receives a scalar value $Q(x)$ from a reward model that evaluates the reasoning process. The reward model input consists of the problem statement and the sequence of reasoning steps along the path from the root to $x$, formatted as alternating user–assistant turns in a conversation history (Appendix E).

**Models and main datasets.** We evaluate three widely available open-weight LLMs: Llama-3.1-8B-Instruct (Grattafiori et al., 2024), Qwen2.5-7B-Instruct (Qwen et al., 2025), and Qwen3-32B (Team, 2025). For Qwen3-32B, we disable its reasoning mode, as it consumes a substantial portion of the output-token budget. This regime reflects *resource-constrained settings* where scaling model size is often infeasible and test-time scaling is the more realistic lever. As the reward model, we use GenPRM-7B (Zhao et al., 2025; Liu et al., 2025a). Our main experiments use the mathematical reasoning benchmarks MATH500 (Lightman et al., 2024) and AIME24/25 (Maxwell-Jia, 2024; OpenCompass, 2025).

**Additional physics evaluation.** As a complementary check beyond mathematical reasoning, we also evaluate BG-MCTS on UGPhysics (Xu et al., 2025), using Qwen2.5-7B-Instruct and Qwen3-32B. This evaluation is intended as an additional check of the empirical trend beyond the mathematical benchmarks used in our main experiments.

**Baselines.** We evaluate the following baselines: Repeated Sampling (a representative parallel-scaling baseline) and Sequential Refinement (a representative sequential-scaling baseline), along with hybrid tree-search baselines (MCTS, AB-MCTS-M (Inoue et al., 2025), and LiteSearch (Wang et al., 2024a), which reduces cost via early stopping). Unless otherwise specified, tree-search methods use Sequential generation as the node unit; we use Full generation for the simplest width/depth baselines (Repeated Sampling and Sequential Refinement), and for AB-MCTS-M to match the setting evaluated in the original paper. We disable greedy pre-generation in LiteSearch to align token counting, and otherwise follow method-specific hyperparameters from the original papers. See more details in Appendix B.

**BG-MCTS hyperparameters.** Unless otherwise stated, BG-MCTS expands $k = 2$ children when a leaf node is selected and uses the exploration constant $c = \sqrt{2}$. We set the completion-bias coefficient in Eq. 5 and the widening coefficient in Eq. 6 to $\kappa = 1$ and $\lambda = 1$, respectively.

### 4.2. Main Results on Mathematical Reasoning

**Fixed-budget accuracy.** Table 1 reports accuracy under fixed token budgets. Across models, benchmarks, and budgets, BG-MCTS delivers the strongest overall performance, consistently outperforming budget-agnostic tree-search baselines and surpassing strong sampling-based baselines in most settings. Figure 3 further shows that BG-

*Table 1.* **Accuracy averaged over three trials**. Greedy: no search; subscript "Full": full-generation nodes. **Bold**/underline: best/second-best per budget $B$. BG-MCTS consistently outperforms baselines under fixed budgets. †Details of the aggregation methods for LiteSearch are provided in Appendix C.

| Methods \ Budget $B$ | MATH500 (Lv.5) | | | | AIME24/25 | | | |
|---|---|---|---|---|---|---|---|---|
| | 10K | 20K | 30K | .avg | 10K | 20K | 30K | .avg |
| Llama-3.1-8B-Instruct | | | | | | | | |
| - Greedy | .224 | .224 | .224 | .224 | .033 | .033 | .033 | .033 |
| - Refinement$_{\text{Full}}$ | .249 | .246 | .241 | .245 | .033 | .028 | .028 | .030 |
| - Repeated$_{\text{Full}}$ | **.393** | .438 | **.450** | .427 | .039 | .050 | .056 | .048 |
| - AB-MCTS-M$_{\text{Full}}$ | .311 | .323 | .343 | .326 | .050 | .050 | .050 | .050 |
| - AB-MCTS-M | .124 | .179 | .209 | .171 | .000 | .011 | .011 | .007 |
| - MCTS | .333 | .406 | .430 | .390 | .039 | .072 | .089 | .067 |
| - LiteSearch-Incre.† | .249 | .291 | .304 | .281 | .033 | .039 | .056 | .043 |
| - LiteSearch-Batch.† | .236 | .271 | .281 | .263 | .033 | .044 | .039 | .039 |
| - **BG-MCTS** (ours) | **.393** | **.465** | .443 | **.434** | **.072** | **.083** | **.106** | **.087** |
| Qwen2.5-7B-Instruct | | | | | | | | |
| - Greedy | .493 | .493 | .493 | .493 | .100 | .100 | .100 | .100 |
| - Refinement$_{\text{Full}}$ | .498 | .493 | .490 | .493 | .094 | .083 | .089 | .089 |
| - Repeated$_{\text{Full}}$ | .632 | .664 | .702 | .666 | **.156** | .161 | **.183** | .167 |
| - AB-MCTS-M$_{\text{Full}}$ | .629 | .657 | .659 | .648 | .144 | .150 | .156 | .150 |
| - AB-MCTS-M | .433 | .542 | .600 | .525 | .072 | .128 | .139 | .113 |
| - MCTS | .619 | .657 | .659 | .645 | **.156** | .167 | .167 | .163 |
| - LiteSearch-Incre.† | .488 | .505 | .510 | .501 | .089 | .083 | .083 | .085 |
| - LiteSearch-Batch† | .527 | .550 | .557 | .545 | .078 | .083 | .083 | .082 |
| - **BG-MCTS** (ours) | **.662** | **.699** | **.711** | **.691** | **.156** | **.189** | **.183** | **.176** |
| Qwen3-32B | | | | | | | | |
| - Greedy | .731 | .731 | .731 | .731 | .267 | .267 | .267 | .267 |
| - Refinement$_{\text{Full}}$ | .749 | .744 | .739 | .744 | .228 | .228 | .228 | .228 |
| - Repeated$_{\text{Full}}$ | .766 | .761 | .766 | .765 | .250 | .300 | .306 | .285 |
| - AB-MCTS-M$_{\text{Full}}$ | .749 | .771 | .766 | .762 | **.272** | .267 | .272 | .270 |
| - AB-MCTS-M | .565 | .679 | .711 | .652 | .206 | .222 | .261 | .230 |
| - MCTS | .704 | .779 | .791 | .758 | .244 | .289 | .300 | .278 |
| - LiteSearch-Incre.† | .729 | .736 | .746 | .737 | .256 | .267 | .267 | .263 |
| - LiteSearch-Batch† | .711 | .719 | .726 | .719 | .206 | .228 | .233 | .222 |
| - **BG-MCTS** (ours) | **.784** | **.801** | **.809** | **.798** | .267 | **.317** | **.350** | **.311** |

MCTS reaches its peak performance near budget exhaustion, consistent with its intended budget-aligned wide-to-deep behavior (§ 3). Overall, these results support our core premise: under a fixed budget, conditioning tree-search decisions on the *remaining* budget is more effective than treating the budget merely as a stopping condition.

**Baseline trends.** AB-MCTS-M is less competitive in our fixed-budget setting than simple sampling/refinement baselines such as Repeated Sampling. This appears to reflect a mismatch in operating regime: AB-MCTS-M was reported to be most effective with richer feedback and larger compute (Inoue et al., 2025), whereas our focus is budget-constrained inference with scalar reward-model scores and relatively small token limits. LiteSearch (Wang et al., 2024a) reliably reduces token usage, but often stops prematurely, leaving part of the budget unused and yielding lower final accuracy (Table 1 and Fig. 3).

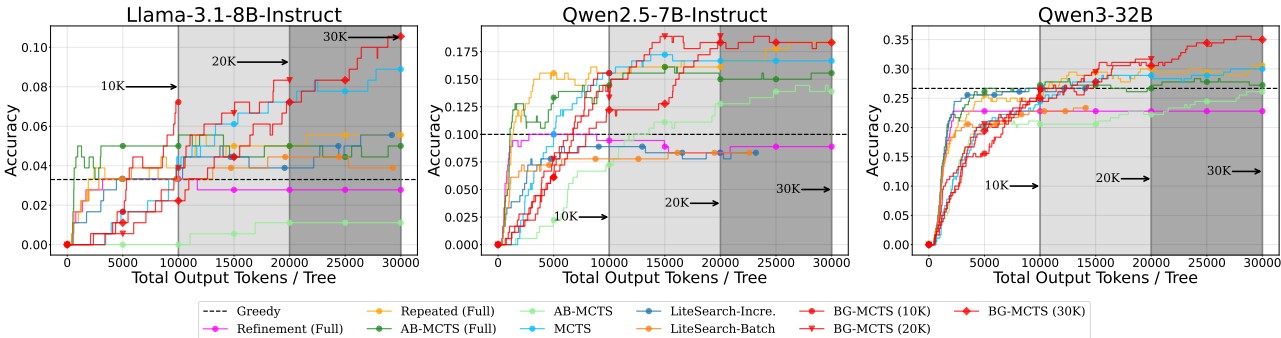

*Figure 3.* **Accuracy vs. consumed output tokens on AIME24/25.** Vertical markers indicate the fixed budgets $B \in \{10K, 20K, 30K\}$. BG-MCTS continues improving toward budget exhaustion and outperforms budget-agnostic baselines at the budget limits. LiteSearch aggregation details and additional plots are provided in Appendices C and F.

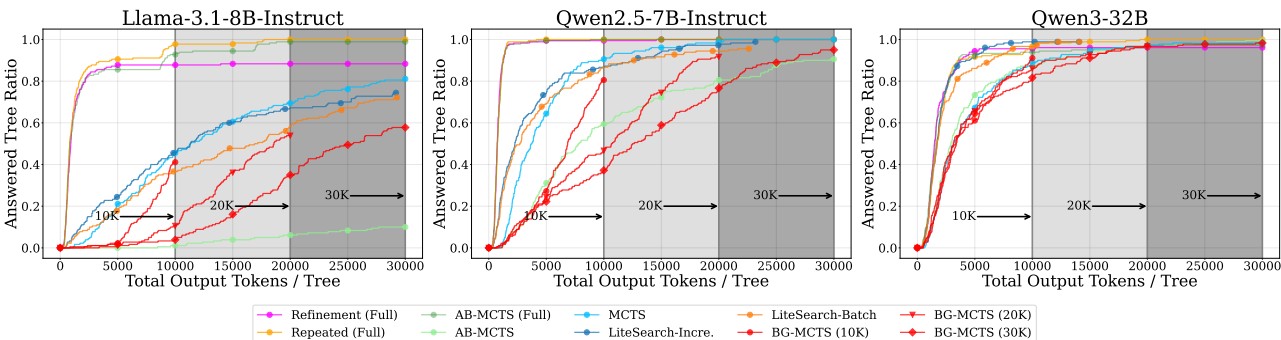

*Figure 4.* **Answered-tree rate vs. consumed output tokens on AIME24/25.** We report the fraction of search trees containing at least one answered node. BG-MCTS increases this rate toward budget exhaustion, consistent with its budget-aware shift toward completion. LiteSearch aggregation details and additional plots are provided in Appendices C and F.

*Table 2.* **Accuracy on MATH500 Lv.5 averaged over two runs.** Each column disables one component: Explore annealing (Eq. 3), Exploit shaping (Eq. 5), or Widen annealing (Eq. 6). Red cells indicate drops from full BG-MCTS; **bold red** indicates performance below MCTS baseline. Full BG-MCTS performs best overall.

| Eq. 3 | Eq. 5 | Eq. 6 | Llama-3.1-8B-Inst. | | | Qwen2.5-7B-Inst. | | |
|---|---|---|---|---|---|---|---|---|
| **Explore annealing** | **Exploit shaping** | **Widen annealing** | **10K** | **20K** | **30K** | **10K** | **20K** | **30K** |
| - | - | - | .333 | .406 | .430 | .619 | .657 | .659 |
| ✓ | - | - | .377 | .478 | **.418** | .649 | **.653** | .679 |
| - | ✓ | - | .340 | .418 | **.425** | .646 | .664 | .660 |
| - | - | ✓ | **.310** | **.392** | .433 | .642 | **.526** | .664 |
| ✓ | ✓ | - | .407 | .470 | .433 | .642 | .675 | **.657** |
| ✓ | - | ✓ | .433 | .425 | .485 | .631 | .690 | .720 |
| - | ✓ | ✓ | **.373** | **.403** | **.429** | **.605** | .660 | .679 |
| ✓ | ✓ | ✓ | .393 | .465 | .443 | .662 | .699 | .711 |

**Ablation.** Table 2 ablates the three main components of BG-MCTS: budget-annealed exploration (Eq. 3), the completion bias (Eq. 5), and budget-guided widening (Eq. 6). Removing any one component generally degrades performance, and no ablated variant consistently matches the full method across budgets. This suggests that the three components address complementary aspects of fixed-budget search.

### 4.3. Additional Physics Evaluation

As a complementary check beyond our main mathematical reasoning benchmarks, we evaluate BG-MCTS on a physics reasoning benchmark, UGPhysics (Xu et al., 2025). It is intended to assess whether the empirical trends observed on math reasoning also appear in a physics reasoning setting.

**Rollout-based node evaluation.** Since no off-the-shelf reward model is available for evaluating partial physics solutions, we use a reference-answer-based rollout evaluator. For each newly expanded non-terminal node $x$, we perform $n = 5$ rollouts until final answers are generated, and set $Q(x)$ to the fraction of rollouts that match the reference answer. Terminal nodes that already contain a final answer are assigned binary rewards. Because terminal-node scores are therefore not directly comparable to rollout-based non-terminal scores, we select the representative answer by majority voting over generated final answers. We also report the results of LiteSearch (Wang et al., 2024a) without its early-stopping, since binary terminal rewards do not provide a meaningful confidence threshold in this setting.

**Models and datasets.** We evaluate two widely available open-weight Qwen-family LLMs, Qwen2.5-7B-

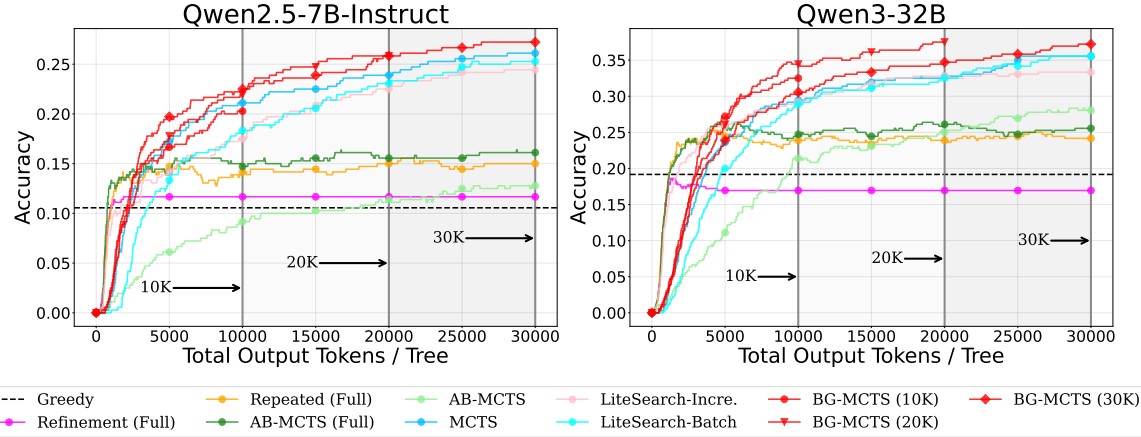

*Figure 5.* **Accuracy vs. consumed output tokens on UGPhysics.** BG-MCTS improves toward budget exhaustion and outperforms budget-agnostic baselines at the budget limits $B \in \{10K, 20K, 30K\}$.

*Table 3.* **Accuracy on UGPhysics (Xu et al., 2025) averaged over three runs.** Greedy: no search; subscript "Full": full-generation nodes. **Bold**/underline: best/second-best per budget $B$. BG-MCTS consistently outperforms baselines under fixed budgets.

| | UGPhysics | | | | | | | |
| | Qwen2.5-7B-Inst. | | | | Qwen3-32B | | | |
| Methods \ Budget $B$ | 10K | 20K | 30K | .avg | 10K | 20K | 30K | .avg |
|---|---|---|---|---|---|---|---|---|
| - Greedy | .106 | .106 | .106 | .106 | .192 | .192 | .192 | .192 |
| - Refinement$_{Full}$ | .117 | .117 | .117 | .117 | .169 | .169 | .169 | .169 |
| - Repeated$_{Full}$ | .142 | .150 | .150 | .147 | .239 | .239 | .242 | .240 |
| - AB-MCTS-M$_{Full}$ | .147 | .156 | .161 | .155 | .247 | .261 | .256 | .255 |
| - AB-MCTS-M | .092 | .114 | .128 | .111 | .214 | .250 | .281 | .248 |
| - MCTS | **.211** | .239 | .261 | .237 | .294 | .328 | .356 | .326 |
| - LiteSearch-Incre.[†] | .175 | .225 | .244 | .215 | .292 | .328 | .333 | .318 |
| - LiteSearch-Batch[†] | .183 | .231 | .253 | .222 | .289 | .325 | .356 | .323 |
| - **BG-MCTS** (ours) | .203 | **.258** | **.272** | **.246** | **.325** | **.375** | **.372** | **.357** |

*Table 4.* **Answered-node statistics on AIME24/25 with Llama-3.1-8B-Instruct.** We report the number of answered nodes and correct answered nodes (Cor.) for each budget $B$. **Bold**/underline: best/second-best. BG-MCTS produces fewer but higher-quality answered nodes. Complete statistics across models, datasets, and budgets are provided in Appendices C and L.

| Budget $B$ | 10K | | 20K | | 30K | |
| Methods | Total | Cor. | Total | Cor. | Total | Cor. |
|---|---|---|---|---|---|---|
| Refinement$_{Full}$ | 277.0 | 12.0 | 390.3 | 14.3 | 480.3 | 16.3 |
| Repeated$_{Full}$ | 365.3 | 14.3 | 704.3 | 32.3 | **1038.7** | 48.0 |
| AB-MCTS-M$_{Full}$ | 317.0 | 13.7 | 543.0 | 20.3 | 739.0 | 25.7 |
| AB-MCTS-M | 0.7 | 0.0 | 5.0 | 1.0 | 8.7 | 1.3 |
| MCTS | 188.3 | 21.3 | 492.3 | 60.0 | 916.3 | 143.7 |
| LiteSearch-Incre.[†] | **503.7** | 16.0 | **838.0** | 43.0 | 976.3 | 51.3 |
| LiteSearch-Batch[†] | 416.0 | 15.3 | 645.0 | 17.3 | 849.0 | 18.0 |
| **BG-MCTS** (ours) | 198.7 | **103.0** | 636.0 | **293.7** | 899.0 | **413.7** |

Instruct (Qwen et al., 2025) and Qwen3-32B (Team, 2025), on UGPhysics (Xu et al., 2025). These models provide two different model scales while keeping the model family fixed, allowing us to assess the physics setting without introducing additional cross-family variation. We randomly sample 120 questions from the `Math Derivation`, `Laws Application`, and `Practical Application` categories, focusing on reasoning-intensive tasks (Appendix D).

**Results.** Table 3 reports accuracy under fixed output-token budgets on UGPhysics. Under this rollout-based physics protocol, BG-MCTS achieves the best or near-best accuracy in nearly all settings, consistently improving over budget-agnostic tree-search baselines. Figure 5 further shows that BG-MCTS reaches its peak performance near budget exhaustion, matching the intended budget-aligned behavior. Overall, these results provide complementary evidence that the empirical trends observed on mathematical reasoning can also appear in a physics reasoning setting.

## 5. Analysis

### 5.1. Answer-Reach Rate over Budget

**Tree level.** Figure 4 plots the *tree-level* answer reach rate, defined as the fraction of instances whose search tree contains at least one answered node, as the search consumes its budget. Baselines tend to reach answered nodes early. In contrast, BG-MCTS exhibits a pronounced late-stage rise, suggesting that it delays answer commitment early and increases completion pressure near budget exhaustion. Notably, BG-MCTS attains higher final accuracy despite a lower tree-level reach rate at exhaustion (Tab. 1; Fig. 3).

**Node level.** To explain this gap, we further analyze *node-level* behavior. Table 4 reports the total number of answered and correct answered nodes. BG-MCTS produces more correct answered nodes even when its tree-level reach rate is lower, indicating that it trades answer coverage for higher-quality completed candidates.

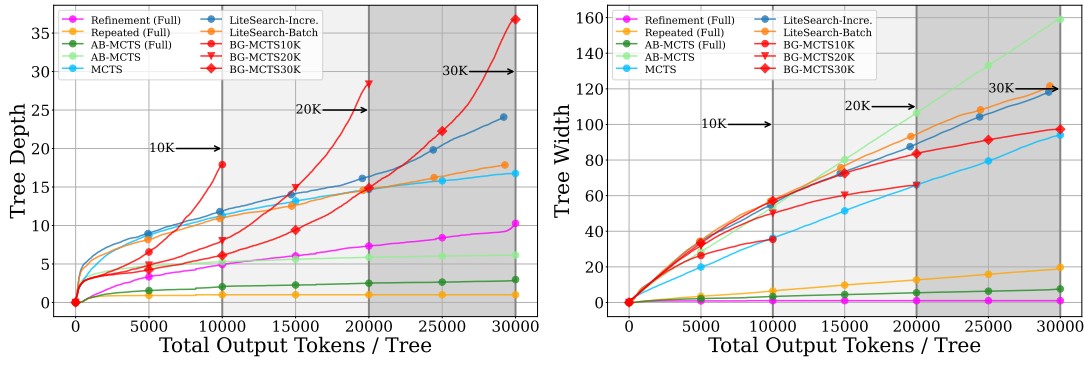

*(a)* Average maximum depth of search tree      *(b)* Average maximum width of search tree

*Figure 6.* **Average maximum depth and width of search trees on AIME24/25 with Llama-3.1-8B-Instruct.** BG-MCTS maintains broader exploration early in the search, then increasingly shifts toward deeper branches as the remaining token budget decreases. LiteSearch aggregation details are provided in Appendix C; full results across models and benchmarks are provided in Appendix F.

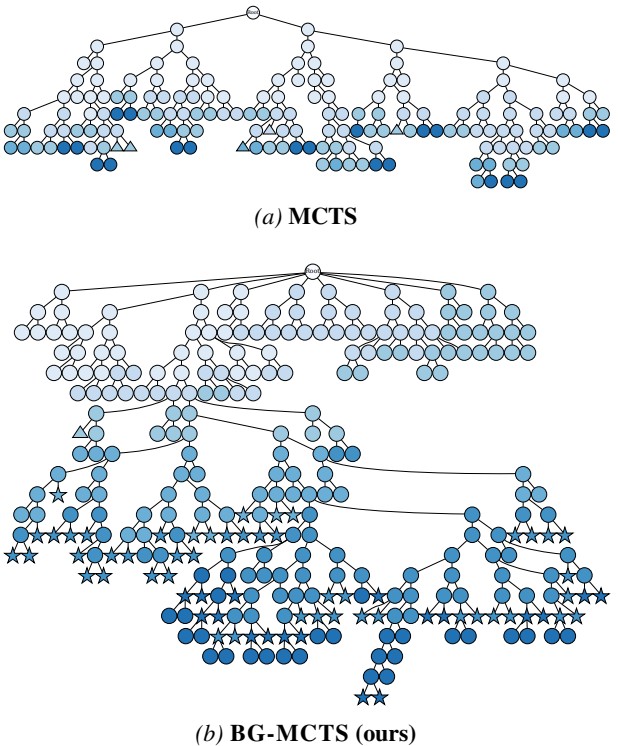

*(a)* **MCTS**

*(b)* **BG-MCTS (ours)**

*Figure 7.* **Tree examples of MCTS vs. BG-MCTS** (Llama-3.1-8B-Instruct, MATH500 Level 5, budget 20K). Stars and triangles denote correct and incorrect nodes; color intensity reflects expansion order (darker = later). BG-MCTS adaptively shifts to depth-first as budget depletes. Details in Appendix M.

## 5.2. Depth–Width Trade-off Analysis

Figures 6a and 6b report the average maximum depth and maximum width of the full search tree produced by each method. These two statistics characterize how each method allocates its fixed budget between deepening existing trajectories and widening the search frontier. The depth analysis

(Fig. 6a) shows that BG-MCTS increasingly favors deeper branches as the search consumes its token budget, reflecting a late-stage shift toward refining promising candidates. The width analysis (Fig. 6b) provides a complementary view: BG-MCTS maintains sufficient breadth when the remaining budget is large, but progressively suppresses further widening as the budget decreases. This pattern contrasts with budget-agnostic search policies, whose depth–width behavior is not explicitly adjusted according to the remaining budget. Together, these results show that BG-MCTS realizes the intended depth–width trade-off, moving from broad exploration to deeper refinement under a fixed budget.

## 5.3. Qualitative analysis

Figure 7 visualizes the search trees produced by MCTS and BG-MCTS under the same fixed budget ($B$=20K) on a MATH500 Level-5 problem using Llama-3.1-8B-Instruct. Nodes marked with a star ($\star$) and a triangle ($\triangle$) denote correct and incorrect final answers, respectively; all other nodes are intermediate process nodes. Darker blue indicates nodes expanded later in the search, when less budget remains.

Standard MCTS (Fig. 7a) treats the budget mainly as a stopping condition, so its expansion pattern remains largely unchanged as the search consumes the budget. As a result, it continues widening at shallow depths even late in the run, leaving insufficient budget to deepen and complete promising branches. In this example, MCTS broadens the tree but never reaches a correct final answer.

In contrast, BG-MCTS (Fig. 7b) exhibits the intended budget-aligned wide-to-deep behavior: broad early exploration followed by increasingly focused refinement within a small number of promising subtrees. This late-stage focus enables search to complete a promising trajectory and reach a correct final answer under the same budget, illustrating the effect of budget-conditioned selection and widening.

# 6. Discussion

**Reward model.** Our experiments in Sec. 4 use GenPRM-7B (Zhao et al., 2025; Liu et al., 2025a) as the process reward model (PRM) for node evaluation. As shown in Table 10 in Appendix K, its scores are often highly saturated and near-binary in our setting, yielding a weak reward signal for fine-grained node selection. In such a regime, tree search is less able to rank promising partial trajectories precisely and relies more on filtering out clearly poor ones. Because all compared methods use the same evaluator, this does not affect the fairness of our empirical comparisons, but it does limit the granularity of the search signal. Combined with the gains in Table 1, this suggests that budget-aware selection and widening remain useful even with an imperfect PRM. Better-calibrated reward models or process evaluators that more reliably distinguish partial progress may further strengthen budget-constrained tree-search decoding.

**Token budget as a decoding interface.** BG-MCTS treats fixed-budget inference by conditioning search behavior on the remaining budget, rather than using the budget only as a stopping criterion. This perspective is particularly relevant for API-served frontier models, where inference is often token-metered and per-query spending must be predictable.[3] More broadly, budget-conditioned decoding provides a simple interface for *predictable* test-time scaling: users specify a token budget, and the decoding policy adapts its exploration-to-completion schedule within the resulting cost envelope.

**Budgeted data synthesis.** Our analysis (Sec. 5.1 and 5.2) suggests a "fewer-instances, better-answers" regime: BG-MCTS may reach final answers on fewer problem instances, but the answered nodes it produces have higher precision and include more correct candidates when multiple outputs are retained. This behavior is well suited to data synthesis, where the quality of generated reasoning traces can matter more than raw coverage; prior work similarly suggests that smaller, cleaner sets of traces can support efficient learning (Gunasekar et al., 2023; Chunting et al., 2023; Muennighoff et al., 2025). When multiple trajectories are retained for the same prompt, the successful and unsuccessful candidates can also serve as offline comparison data for preference optimization, beyond supervised fine-tuning. Finally, since BG-MCTS operates under a prescribed per-instance budget, it provides a predictable cost profile while steering synthesis toward more reliable completed traces, simplifying large-scale generation, filtering, and iteration.

---

[3]API pricing is often model-dependent and token-metered. As an example, for frontier models, output tokens can cost on the order of tens of dollars per million tokens. See `https://open ai.com/api/pricing/`.

# 7. Related Work

**Test-time scaling.** Test-time scaling improves generation quality of LLMs by allocating additional compute at *inference* time without updating model parameters. Prior work broadly falls into three directions: (i) *parallel* scaling, which samples multiple candidates and aggregates or selects among them (Wang et al., 2023; Brown et al., 2024; Snell et al., 2024; Lightman et al., 2024; Komiyama et al., 2026), (ii) *sequential* scaling, which iteratively refines a single trajectory based on intermediate states (Madaan et al., 2023; Yao et al., 2023b; Muennighoff et al., 2025), and (iii) *hybrid* scaling, which combines both by maintaining multiple trajectories and selecting among them through tree search (Tian et al., 2024; Zhou et al., 2024; Inoue et al., 2025; Zheng et al., 2025). Hybrid approaches often use MCTS/PUCT-style selection (Silver et al., 2017), but are typically budget-agnostic: the search policy is fixed and the budget is used mainly as a termination condition.

**Reducing search costs.** LiteSearch (Wang et al., 2024a) reduces token usage through pruning and early stopping. While LiteSearch aims to lower inference cost, our goal is to allocate a *given* fixed budget more effectively. As a result, early stopping can be suboptimal in our setting, since it may terminate search before the remaining budget is used for further refinement.

**Budget-aware prompting for agents.** Liu et al. (2025b) propose BATS, a budget-aware prompting method for tool-augmented search agents, where remaining per-tool-call budgets are exposed in the prompt to guide planning and verification in a sequential ReAct-style loop (Yao et al., 2023b). In contrast, BG-MCTS targets *tree-search decoding* under a fixed output-token budget: the remaining budget ratio $\rho$ is built directly into the MCTS selection and widening rules, inducing an algorithmic shift from exploration to completion rather than relying on prompt-level adaptation.

# 8. Conclusion

We presented Budget-Guided MCTS (BG-MCTS), a budget-aware tree-search decoding method for LLMs that conditions both selection and widening on the remaining token budget. Across mathematical reasoning benchmarks, with an additional complementary evaluation on UGPhysics, BG-MCTS improves fixed-budget inference by inducing a budget-aligned search pattern: broad exploration early in the search, followed by refinement and completion as the remaining budget decreases.

Future work includes extending the budget modeling to account for input tokens and reward-model computation, as well as studying budget allocation across multiple reward models and broader accuracy–cost trade-offs.

## Acknowledgment

This work was partially supported by the "R&D Hub Aimed at Ensuring Transparency and Reliability of Generative AI Models" project of the Ministry of Education, Culture, Sports, Science and Technology (MEXT). This work was partially supported by JST K Program Japan Grant Number JPMJKP24C3 and JSPS KAKENHI Grant Number 25H01137. This study was carried out using the TSUBAME4.0 supercomputer at Institute of Science Tokyo.

## Impact Statement

This paper presents work whose goal is to advance the field of machine learning. There are many potential societal consequences of our work, none of which we feel must be specifically highlighted here.

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

## A. Methodology for Answer Extraction

We identify a node as an answer node if it satisfies either of the following conditions. First, the generated text matches the regular expression pattern `r"answer is(.*)\\boxed\{.*?\}"`. Second, generation terminates naturally, rather than being forcibly truncated by a maximum token limit or by the delimiter `\nStep`.

## B. Hyperparameter Search for Baseline MCTS

Because the hyperparameters for the baseline Monte Carlo Tree Search (MCTS) vary across prior studies, we perform a small hyperparameter search for the baseline MCTS. Specifically, using Llama-3.1-8B-Instruct on MATH500 Level-5 problems, we search over the exploration constant $c \in \{\sqrt{0.1}, \sqrt{1.0}, \sqrt{2.0}\}$ in the PUCT score (Eq. 1) and the number of children expanded at each selected leaf, chosen from $\{2, 3, 5\}$. The search is conducted under a fixed token budget of $B = 15$k. As a result, we set $c = \sqrt{2}$ and expand two children per selected leaf, which achieved the highest accuracy.

## C. Budget Accounting for LiteSearch

LiteSearch (Wang et al., 2024a) includes an early-stopping mechanism: when a node containing a final answer receives a reward-model score above a threshold $\epsilon$ (we use $\epsilon = 0.9$), the search terminates before exhausting the nominal budget. Thus, for a nominal budget $B$, the actual consumed budget $B_{\text{actual}}$ may satisfy $B_{\text{actual}} < B$. We therefore use two budget-accounting schemes: one for fixed-budget tables and one for budget-trajectory plots.

**Per-instance fixed-budget evaluation (tables).** For each budget $B \in \{10\text{K}, 20\text{K}, 30\text{K}\}$, each problem instance is run independently with a maximum output-token budget of $B$. If LiteSearch stops early at $B' < B$, we use the tree state at early stopping as the result for budget $B$. This matches a deployment setting in which token usage is avoided once the stopping criterion is satisfied, while all methods are evaluated under the same nominal per-instance budget.

**Budget-redistributed aggregation (figures).** For plots as a function of consumed tokens, we additionally use a budget-redistributed aggregation scheme to visualize the dataset-level effect of early stopping. At a nominal budget $B$, let $\mathcal{I}_{\text{stop}}$ be the set of early-stopped instances and $\mathcal{I}_{\text{act}}$ the remaining active instances. The total saved budget is $R = \sum_{i \in \mathcal{I}_{\text{stop}}}(B - B_{\text{actual},i})$, which is redistributed uniformly over active instances by evaluating them at $B_{\text{adj}} = B + R/|\mathcal{I}_{\text{act}}|$. We cap $B_{\text{adj}}$ at $B_{\text{max}} = 30\text{K}$ for every instance. This aggregation is used only for visualization: it shows how LiteSearch trades accuracy against consumed tokens when early-stopped instances save budget that could otherwise be spent on unfinished instances.

## D. UGPhysics Sampling Procedure

UGPhysics (Xu et al., 2025) consists of physics problems covering 13 subject areas. Within each subject area, problems are categorized into five problem types: `Knowledge Recall`, `Laws Applications`, `Math Derivation`, `Practical Application`, and `Others`. Since our focus is on reasoning-intensive inference tasks, in our additional evaluation for physics reasoning (Sec. 4.3), we excluded `Knowledge Recall` and `Others` categories, and we randomly sampled 40 questions from each of the three remaining categories, resulting in a total of 120 questions (Table 5). The sampling is performed across all 13 subject areas without stratifying by subject area.

*Table 5.* **Distribution of UGPhysics questions used in our experiments (Sec. 4.3).**

| Subjects / `Problem types` | `Laws Application` | `Math Derivation` | `Practical Application` | Total |
|---|---|---|---|---|
| Atomic Physics | 8 | 0 | 13 | 21 |
| Classical Electromagnetism | 4 | 3 | 0 | 7 |
| Classical Mechanics | 9 | 7 | 7 | 23 |
| Electrodynamics | 0 | 3 | 0 | 3 |
| Geometrical Optics | 1 | 1 | 1 | 3 |
| Quantum Mechanics | 4 | 9 | 2 | 15 |
| Relativity | 0 | 3 | 0 | 3 |
| Semiconductor Physics | 1 | 1 | 1 | 3 |
| Solid-State Physics | 3 | 1 | 7 | 11 |
| Statistical Mechanics | 4 | 4 | 1 | 9 |
| Theoretical Mechanics | 1 | 5 | 0 | 6 |
| Thermodynamics | 1 | 2 | 0 | 3 |
| Wave Optics | 4 | 1 | 8 | 13 |
| **Total** | **40** | **40** | **40** | **120** |

# E. Prompt

**Input to the LLM.** Figure 8 shows the chat-format prompt template used for Llama-3.1-8B-Instruct and Qwen2.5-7B-Instruct, for both Sequential and Full generation. The prompt includes few-shot examples that encourage the model to solve the problem step by step. To facilitate step-by-step generation, we fix the beginning of the assistant response to `Step 1:`.

```
Prompt Structure for LLM

{role:  "user", content:  """
Solve the following math problem efficiently and clearly.  The last line of your
response should be of the following format:  'Therefore, the final answer is:
$\\{{boxed{{ANSWER}}$.  I hope it is correct' (without quotes) where ANSWER is the
final number or expression in LaTeX format.  Think step by step before answering.
Example:
Example Problem:
Natalia sold clips to 48 of her friends in April, and then she sold half as many
clips in May.  How many clips did Natalia sell altogether in April and May?
Example Solution:

Step 2:  In May, she sold half as many clips as in April.  Half of 48 is 48 / 2 = 24
clips.

Step 3:  To find the total number of clips sold in April and May, add the number of
clips sold in each month:  48 + 24 = 72.

Step 4:  Therefore the final answer is:  $\\boxed{{72}}$.  I hope it is correct.

Now, solve the following question:  {problem}
"""},
{role:  "assistant", content:  "Step 1:"}
```

*Figure 8.* **Prompt template in a chat format for LLMs (Llama-3.1-8B-Instruct and Qwen2.5-7B-Instruct)**.

**Input to the PRM.** Figure 9 shows the chat-format template used for GenPRM-7B (Zhao et al., 2025; Liu et al., 2025a). For GenPRM-based node evaluation, we provide the full reasoning path from the root to the current node, rather than only the newly generated step. Concretely, the prompt contains the problem statement, the sequence of ancestor reasoning steps, the GenPRM-generated judgments for previous steps, and the current step to be evaluated.

For Full generation baselines, a single node may contain an entire multi-step solution. We therefore decompose the generated solution into individual steps and apply the same step-wise GenPRM evaluation protocol to each prefix. We use the GenPRM score assigned to the final step as the node score of the completed solution.

```
Prompt Structure for GenPRM-7B

{role:  "system", content:  "You are a math teacher.  Your task is to review and
critique the paragraphs in solution step by step."},
{role:  "user", content:  "Question: {problem}\n\nStep 1:  ..."},
{role:  "assistant", content:  "<analyze>\nLet's analyze the Paragraph 1 step by
step:  .....  </analyze>\n<output>\n**Judgement**:  $\\boxed{Yes}$ \n</output>\n"}
{role:  "user", content:  "Step 2:  ... "}
{role:  "assistant", content:  "<analyze>\nLet's analyze the Paragraph 2 step by
step:  .....  </analyze>\n<output>\n**Judgement**:  $\\boxed{Yes}$ \n</output>\n"}
{role:  "user", content:  "Step 3:  ... "}
{role:  "assistant", content:  "<analyze>\nLet's analyze the Paragraph 3 step by
step:  .....  </analyze>\n<output>\n**Judgement**:  $\\boxed{No}$ \n</output>\n"}
{role:  "user", content:  "Step 4:  ... "}
```

*Figure 9.* **Prompt template in a chat format for GenPRM-7B**.

# F. Additional Results

## F.1. Mathematical Reasoning

**Accuracy over consumed budget.** Figure 10 shows solution accuracy as a function of consumed output tokens on MATH500 Level 5. BG-MCTS reaches its peak performance near budget exhaustion.

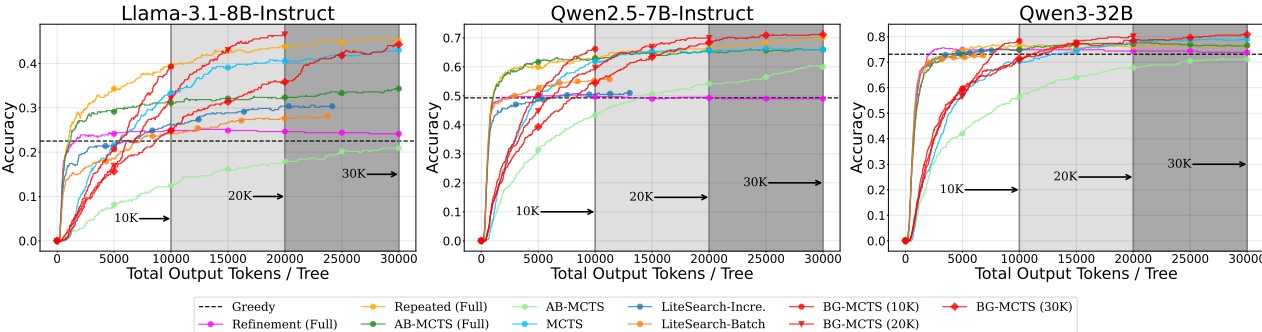

*Figure 10.* **Accuracy vs. consumed output tokens on MATH500 (Level 5).** Vertical markers indicate the fixed budgets $B \in \{10K, 20K, 30K\}$. BG-MCTS continues improving toward budget exhaustion and outperforms budget-agnostic baselines at the budget limits. LiteSearch aggregation details are provided in Appendix C. Results of AIME24/25 are provided in Section 4 (Fig. 3).

**Answer-reach rate.** Figure 11 shows the fraction of search trees containing at least one answered node as a function of consumed output tokens. For BG-MCTS, this fraction increases mainly in the later stage of search, indicating that the method delays answer commitment early and increasingly shifts toward completion as the remaining budget decreases. This behavior is consistent with its budget-guided policy, which moves from broad exploration to deeper refinement as the budget is consumed.

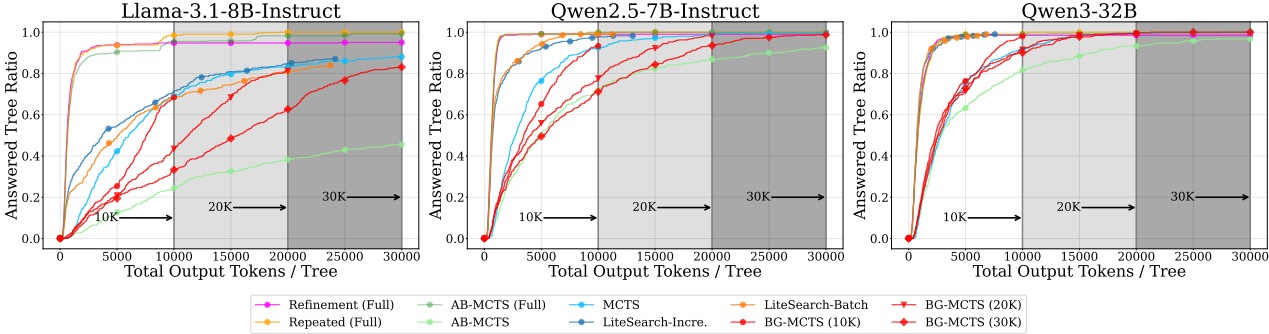

*Figure 11.* **Answered-tree rate vs. consumed output tokens on MATH500 Level 5.** We report the fraction of search trees containing at least one answered node. BG-MCTS increases this rate toward budget exhaustion, consistent with its budget-aware shift toward completion. LiteSearch aggregation details are in Appendix C; AIME24/25 results are shown in Section 5 and Fig. 4.

**Depth and width.** Figures 12 and 13 show the average maximum depth and width of the search tree as functions of consumed output tokens. For BG-MCTS, width expands quickly early on, indicating broad exploration when sufficient budget remains. As the budget is consumed, width growth slows and depth increases more sharply, indicating a shift toward refining existing branches rather than creating new ones. This behavior is consistent with the intended budget-guided transition from breadth to depth.

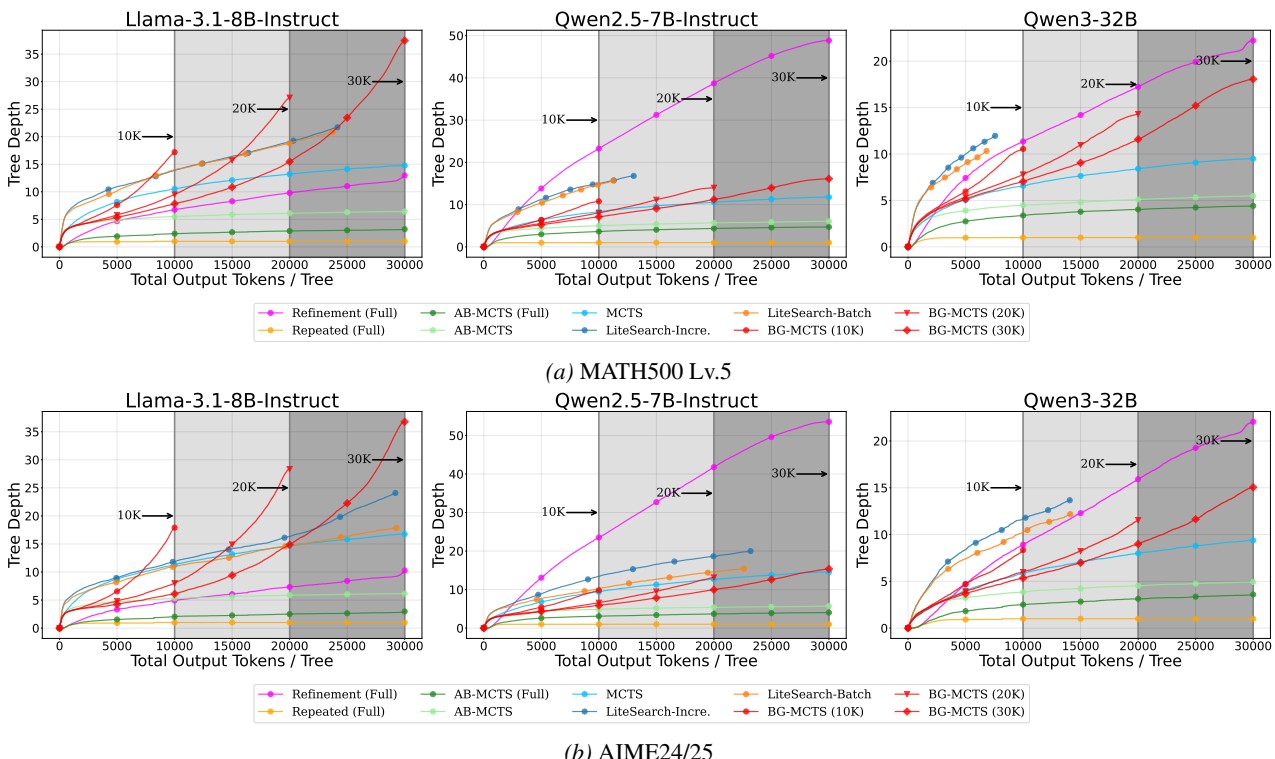

*(a)* MATH500 Lv.5

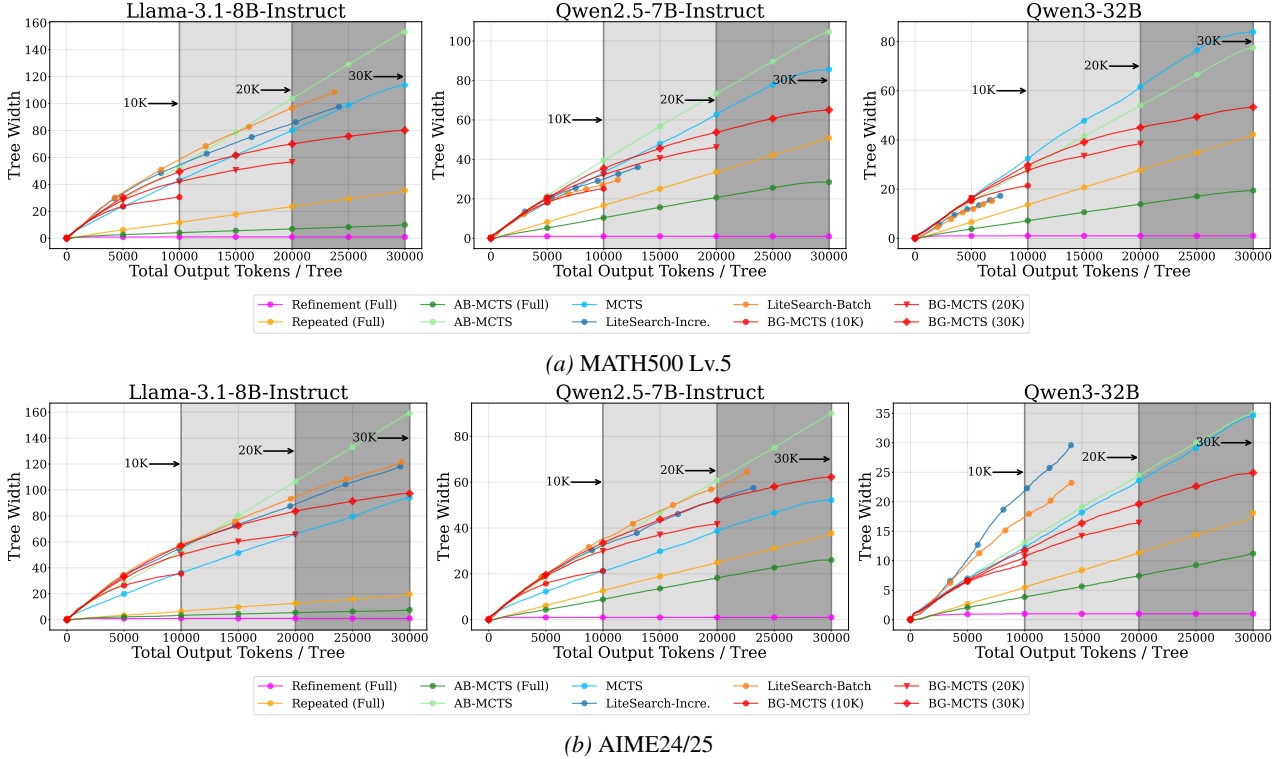

*(b)* AIME24/25

*Figure 12.* **Average maximum depth of search trees.** Results are shown for MATH500 Level 5 (Fig. 12a) and AIME24/25 (Fig. 12b) using Llama-3.1-8B-Instruct, Qwen2.5-7B-Instruct, and Qwen3-32B. LiteSearch aggregation details are provided in Appendix C.

*(a)* MATH500 Lv.5

*(b)* AIME24/25

*Figure 13.* **Average maximum width of search trees on MATH500 Level 5 and AIME24/25.** Results are shown for Llama-3.1-8B-Instruct, Qwen2.5-7B-Instruct, and Qwen3-32B. LiteSearch aggregation details are provided in Appendix C.

### F.2. Physics Reasoning

**Answered-tree rate.** Figure 14 shows the fraction of search trees containing at least one answered node as a function of consumed output tokens.

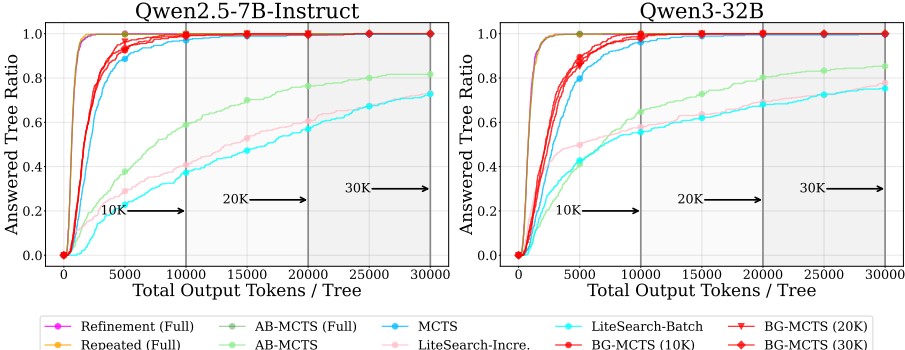

*Figure 14.* **Answered-tree rate vs. consumed output tokens on UGPhysics.** We report the percentage of search trees containing at least one answered node using Qwen2.5-7B-Instruct and Qwen3-32B. LiteSearch aggregation details are provided in Appendix C.

### F.3. Additional Analysis of LiteSearch Baseline

Figure 14 shows that LiteSearch reaches final-answer nodes much less often than other methods on UGPhysics. This contrasts with the mathematical reasoning setting, where LiteSearch tends to reach final answers early in the search (Figs. 11 and 4). We attribute this difference to the reward signals used for node evaluation.

In mathematical reasoning, GenPRM often assigns near-1 scores even to early partial solutions, encouraging LiteSearch to deepen those branches and reach completed answers quickly. In UGPhysics, however, node values are estimated by rollout-based evaluation: at shallow depths, most rollouts do not yet reach correct final answers, so estimated rewards are often close to 0. LiteSearch therefore receives little signal to deepen early branches and instead tends to widen the tree, producing substantially fewer completed answers. This suggests that LiteSearch, and more generally methods that rely heavily on intermediate node scores, can be sensitive to the reward scale and evaluation protocol.

## G. Hyperparameter Sensitivity for BG-MCTS

We examine the sensitivity of BG-MCTS to its main hyperparameters. In the main experiments, we use fixed default values for the exploration constant $c$, the completion-bias coefficient $\kappa$, and the widening coefficient $\lambda$ across tasks and budgets. To assess robustness, we conduct an additional sweep on MATH500 Level 5 using Qwen2.5-7B-Instruct under a fixed 20K-token budget. As shown in Table 6, accuracy ranges from 0.664 to 0.709 across the tested configurations. Several configurations near the default match or exceed the default performance, suggesting that BG-MCTS is not brittle within a moderate neighborhood of the chosen hyperparameters.

*Table 6.* **Hyperparameter sensitivity of BG-MCTS on MATH500 Level 5 with Qwen2.5-7B-Instruct.** Results are obtained under a fixed token budget of $B = 20\text{K}$, with one run per configuration.

| $c$ | $\kappa$ | $\lambda$ | Accuracy |
|-----|----------|-----------|----------|
| $\sqrt{2}$ | 1.0 | 1.0 | .699 |
| $\sqrt{2}$ | 1.0 | 0.5 | .694 |
| $\sqrt{2}$ | 1.0 | 1.5 | .694 |
| $\sqrt{2}$ | 0.5 | 1.0 | .709 |
| $\sqrt{2}$ | 0.5 | 0.5 | .664 |
| $\sqrt{2}$ | 0.5 | 1.5 | .664 |
| 1.0 | 1.0 | 1.0 | .664 |
| 1.0 | 1.0 | 0.5 | .694 |
| 1.0 | 1.0 | 1.5 | .687 |
| 1.0 | 0.5 | 1.0 | .694 |
| 1.0 | 0.5 | 0.5 | .694 |
| 1.0 | 0.5 | 1.5 | .687 |

## H. Effect of Virtual Generation Node

We examine the contribution of the virtual generative node in BG-MCTS. While budget-aware selection can already adjust the preference among existing children as the remaining budget decreases, full BG-MCTS additionally introduces a virtual generative node that makes widening an explicit selectable action. This design allows the search to decide, under the same budget-aware scoring rule, whether to deepen an existing branch or generate a new child from the current node.

Table 7 compares full BG-MCTS with a variant that removes the virtual generative node (Eq. 6), denoted BG-MCTS w/o generative node. This variant keeps the budget-aware selection rule for existing children, but cannot trigger dynamic widening from intermediate nodes. Although it improves over standard MCTS at the 10K-token budget, it is less stable across budgets and does not consistently match full BG-MCTS. These results suggest that the virtual generative node, and the dynamic widening it enables, contributes beyond budget-aware selection alone.

*Table 7.* **Effect of the virtual generative node on MATH500 Level 5 with Qwen2.5-7B-Instruct.** MCTS and full BG-MCTS results are taken from the main experiments, while BG-MCTS (w/o generative node) is based on a single run.

| Method | 10K | 20K | 30K |
|---|---|---|---|
| MCTS | .619 | .657 | .659 |
| BG-MCTS w/o generative node | **.679** | .619 | .672 |
| BG-MCTS | .662 | **.699** | **.711** |

## I. Comparison with Beam-Style Search Baseline

We additionally compare BG-MCTS with a beam-style baseline under the same fixed-budget protocol. We use standard beam search with beam width 5 and branching factor 10, using the same prompt and model as in the main experiments (Sec. 4). Although beam search maintains multiple candidate trajectories, it expands and prunes only the current frontier, rather than revisiting arbitrary internal nodes. In contrast, BG-MCTS can revisit internal nodes and adaptively decide whether to deepen an existing branch or widen the tree based on the remaining budget.

As in Table 8, beam search performs worse than BG-MCTS in this setting. This suggests that fixed-budget tree search benefits from adaptive internal-node selection and budget-guided widening, rather than frontier-only expansion and pruning.

*Table 8.* **Comparison with beam-style search on MATH500 Level 5 using Qwen2.5-7B-Instruct.** Beam Search uses beam width 5 and branching factor 10. The results for BG-MCTS are taken from the main experiments, while Beam Search is based on a single run.

| Method | 10K | 20K | 30K |
|---|---|---|---|
| Beam Search | .328 | .537 | .597 |
| BG-MCTS | **.662** | **.699** | **.711** |

## J. BG-MCTS with Early Stopping

LiteSearch includes an early-stopping mechanism that terminates search once a sufficiently high-scoring final answer is found. This mechanism is enabled in our main LiteSearch baselines; we only disable greedy pre-generation to keep token accounting consistent across methods. Thus, the main experiments already compare BG-MCTS against LiteSearch with early stopping. Here, we additionally apply the same early-stopping criterion to BG-MCTS to separate the effect of early stopping from the effect of budget-guided search.

We conduct this comparison on MATH500 Level 5 using Llama-3.1-8B-Instruct under a fixed 20K-token budget. As shown in Table 9, BG-MCTS with early stopping achieves higher accuracy than both LiteSearch variants while using fewer tokens on average. Specifically, BG-MCTS with early stopping reaches 0.396 accuracy with 12.94K tokens, compared with 0.291/16.40K for LiteSearch-Incre. and 0.271/16.16K for LiteSearch-Batch. Using the full 20K budget, BG-MCTS achieves the highest accuracy overall, reaching 0.465.

These results suggest that early stopping and budget-guided search address complementary aspects of test-time computation. Early stopping reduces token usage once a high-confidence answer is found, whereas BG-MCTS improves how the available budget is allocated during search. When the same early-stopping criterion is applied, BG-MCTS remains substantially stronger than LiteSearch while also using fewer tokens on average.

*Table 9.* **Effect of applying early stopping to BG-MCTS on MATH500 Level 5 with Llama-3.1-8B-Instruct.** All methods are run under a fixed token budget of $B = 20$K. For BG-MCTS with early stopping, we use the same stopping criterion as LiteSearch.

| Method | Accuracy | Avg. output tokens |
|---|---|---|
| LiteSearch-Incre. | .291 | 16.40K |
| LiteSearch-Batch | .271 | 16.16K |
| BG-MCTS w/ early stopping | .396 | 12.94K |
| BG-MCTS | **.465** | 20.00K |

## K. PRM Score Saturation

We additionally inspect the score distribution of GenPRM-7B (Zhao et al., 2025; Liu et al., 2025a), which we use as the process reward model (PRM) for node evaluation. Table 10 reports score percentiles observed during search on MATH500 Level 5 and AIME24/25 under a fixed token budget of $B = 30$K using Qwen2.5-7B-Instruct. The scores are highly saturated and near-binary: many evaluated nodes receive scores close to 1, while lower percentiles can be close to 0. This creates a weak-signal regime in which the PRM provides limited fine-grained discrimination among promising partial trajectories. Since the same PRM is used across methods, this saturation does not affect the fairness of our comparisons, but it limits the granularity of the selection signal available to tree search. Our results therefore suggest that BG-MCTS remains beneficial even with an imperfect PRM, while better-calibrated PRMs may further improve budget-constrained tree-search decoding.

*Table 10.* **Percentiles of GenPRM-7B evaluation scores observed during search.** Scores are collected on MATH500 Level 5 and AIME24/25 under a fixed token budget of $B = 30$K using Qwen2.5-7B-Instruct.

| Task | $p_{10}$ | $p_{25}$ | $p_{50}$ | $p_{75}$ | $p_{90}$ |
|---|---|---|---|---|---|
| **MATH500 Level 5** | $2.183 \times 10^{-3}$ | $9.740 \times 10^{-1}$ | $9.967 \times 10^{-1}$ | $9.988 \times 10^{-1}$ | $9.996 \times 10^{-1}$ |
| **AIME24/25** | $1.300 \times 10^{-5}$ | $1.170 \times 10^{-3}$ | $9.399 \times 10^{-1}$ | $9.968 \times 10^{-1}$ | $9.990 \times 10^{-1}$ |

## L. Implications for Data Synthesis

We discuss the implications of our answer-candidate statistics for data synthesis. We do not train downstream models in this work; instead, we analyze the quantity and quality of generated answer candidates under fixed token budgets.

### L.1. Single-solution setting

First, we consider a setting where only one solution is retained for each problem. As shown in Figs. 3 and 11, BG-MCTS may produce answered nodes for fewer problem instances than some baselines. This means that, if a data synthesis pipeline requires exactly one solution per problem, BG-MCTS may provide lower coverage. In contrast, full-generation methods tend to produce a completed response for every input problem, making them advantageous in terms of coverage.

However, coverage alone does not determine the usefulness of synthesized data. Prior work suggests that smaller but higher-quality reasoning traces can be more effective for learning than larger, noisier datasets (Gunasekar et al., 2023; Chunting et al., 2023; Muennighoff et al., 2025). Consistent with this view, Table 1 shows that BG-MCTS achieves higher solution accuracy under fixed budgets. Thus, in a single-solution setting, BG-MCTS trades coverage for higher-quality completed solutions.

### L.2. Multiple-candidate setting

Next, we consider a setting where multiple candidate solutions can be retained for each problem. Table 11 reports the total number of answered nodes, the number of correct answered nodes, and precision, defined as the fraction of answered nodes that are correct. In this setting, BG-MCTS often produces substantially more correct answered nodes while maintaining the highest precision across models, benchmarks, and budgets. This suggests that BG-MCTS is particularly useful when a data synthesis pipeline can retain multiple candidates per problem and filter or rank them afterward. Such candidate pools can also support preference-style data construction: correct and incorrect trajectories, or higher- and lower-scoring candidates for the same prompt naturally form comparison pairs for offline preference optimization.

This behavior follows from the design of BG-MCTS. Early in the search, BG-MCTS allocates budget to broad exploration, while later it concentrates computation on promising subtrees and pushes them toward completion. As a result, BG-MCTS may reach answered nodes on fewer trees, but when it does, it tends to produce multiple high-precision candidates around promising reasoning trajectories. This makes BG-MCTS a natural fit for quality-oriented data synthesis under fixed computational budgets.

**Summary.** These results suggest that BG-MCTS is less suited to maximizing per-problem coverage when only one solution is retained, but well suited to generating high-quality candidate pools when multiple solutions per problem are allowed.

*Table 11.* **Answered-node statistics for data-synthesis analysis.** For each model, benchmark, and budget, we report the total number of answered nodes, the number of correct answered nodes (Cor.), and precision. **Bold**/underline indicate best/second-best. LiteSearch aggregation details are provided in Appendix C.

| Benchmark | MATH500 (Lv.5) | | | | | | | | | AIME24/25 | | | | | | | | |
|---|---|---|---|---|---|---|---|---|---|---|---|---|---|---|---|---|---|---|
| Budget $B$ | 10K | | | 20K | | | 30K | | | 10K | | | 20K | | | 30K | | |
| Method | Total | Cor. | (%) | Total | Cor. | (%) | Total | Cor. | (%) | Total | Cor. | (%) | Total | Cor. | (%) | Total | Cor. | (%) |
| **Llama-3.1-8B-Instruct** | | | | | | | | | | | | | | | | | | |
| - Refinement$_{Full}$ | 866.7 | 203.0 | .234 | 1222.7 | 269.7 | .221 | 1480.3 | 317.3 | .214 | 277.0 | 12.0 | .043 | 390.3 | 14.3 | .037 | 480.3 | 16.3 | .034 |
| - Repeated$_{Full}$ | 1527.7 | 570.3 | .373 | 3038.3 | 1144.0 | .377 | 4493.0 | 1709.7 | .381 | 365.3 | 14.3 | .039 | 704.3 | 32.3 | .046 | **1038.7** | 48.0 | .046 |
| - AB-MCTS-M$_{Full}$ | 1021.0 | 295.3 | .289 | 1748.0 | 501.7 | .287 | 2459.7 | 711.0 | .289 | 317.0 | 13.7 | .043 | 543.0 | 20.3 | .037 | 739.0 | 25.7 | .035 |
| - AB-MCTS-M | 121.3 | 77.0 | .635 | 344.3 | 211.0 | .613 | 597.7 | 366.7 | .613 | 0.7 | 0.0 | .000 | 5.0 | 1.0 | .200 | 8.7 | 1.3 | .154 |
| - MCTS | 1310.0 | 860.0 | .656 | 2821.0 | 1855.7 | .658 | 4129.7 | 2747.3 | .665 | 188.3 | 21.3 | .113 | 492.3 | 60.0 | .122 | 916.3 | 143.7 | .157 |
| - LiteSearch-Incre.[†] | 1609.3 | 306.0 | .190 | 2566.3 | 462.7 | .180 | 2990.3 | 544.7 | .182 | **503.7** | 16.0 | .032 | **838.0** | 43.0 | .051 | 976.3 | 51.3 | .053 |
| - LiteSearch-Batch[†] | 1773.3 | 412.0 | .232 | 2799.7 | 539.7 | .193 | 3197.0 | 590.0 | .185 | 416.0 | 15.3 | .037 | 645.0 | 17.3 | .027 | 849.0 | 18.0 | .021 |
| - BG-MCTS (ours) | **2261.0** | **1736.0** | **.768** | **5810.7** | **4732.3** | **.814** | **8143.3** | **6426.7** | **.789** | 198.7 | **103.0** | **.518** | 636.0 | **293.7** | **.462** | 899.0 | **413.7** | **.460** |
| **Qwen2.5-7B-Instruct** | | | | | | | | | | | | | | | | | | |
| - Refinement$_{Full}$ | 2947.7 | 1429.7 | .485 | 4892.7 | 2288.0 | .468 | 6400.3 | 2918.0 | .456 | 1324.3 | 129.3 | .098 | 2334.7 | 236.3 | .101 | 3145.0 | 324.0 | .103 |
| - Repeated$_{Full}$ | 2194.7 | 1249.0 | .569 | 4425.3 | 2523.7 | .570 | 6664.7 | 3795.7 | .570 | 748.7 | 64.0 | .085 | 1472.0 | 128.7 | .087 | 2191.7 | 191.7 | .087 |
| - AB-MCTS-M$_{Full}$ | 2965.0 | 1759.3 | .593 | 6211.3 | 3702.0 | .596 | 9438.3 | 5686.7 | .603 | 1018.7 | 106.3 | .104 | 2128.7 | 243.3 | .114 | 3234.3 | 380.0 | .117 |
| - AB-MCTS-M | 545.7 | 349.7 | .641 | 1422.3 | 914.3 | .643 | 2523.7 | 1656.0 | .656 | 84.3 | 9.0 | .107 | 249.7 | 35.7 | .143 | 412.3 | 57.3 | .139 |
| - MCTS | 2410.0 | 1700.7 | .706 | 5730.7 | 4002.0 | .698 | 9520.0 | 6706.7 | .704 | 770.7 | 125.3 | .163 | 2071.3 | 354.3 | .171 | 3545.3 | 601.7 | .170 |
| - LiteSearch-Incre.[†] | 3268.0 | 454.7 | .139 | 4709.7 | 535.7 | .114 | 5174.0 | 573.0 | .111 | **2492.7** | 71.3 | .029 | **3912.3** | 73.3 | .019 | **4314.3** | 74.7 | .017 |
| - LiteSearch-Batch[†] | 2849.7 | 456.7 | .160 | 4431.0 | 540.3 | .122 | 4750.3 | 591.7 | .125 | 1672.0 | 25.7 | .015 | 3050.0 | 55.3 | .018 | 3486.7 | 72.3 | .021 |
| - BG-MCTS (ours) | **3338.3** | **2838.3** | **.850** | **7901.7** | **6613.0** | **.837** | **12591.3** | **10570.3** | **.839** | 467.3 | **153.3** | **.328** | 1134.7 | **434.3** | **.383** | 2015.3 | **705.3** | **.350** |
| **Qwen3-32B** | | | | | | | | | | | | | | | | | | |
| - Refinement$_{Full}$ | 1487.3 | 1070.0 | .719 | 2254.3 | 1543.0 | .684 | 2830.0 | 1898.7 | .671 | 517.3 | 115.3 | .224 | 917.7 | 201.3 | .219 | 1246.3 | 262.7 | .211 |
| - Repeated$_{Full}$ | 1800.3 | 1501.3 | .834 | 3654.3 | 3051.7 | .835 | 5511.0 | 4591.7 | .833 | 321.3 | 103.0 | .321 | 656.3 | 216.0 | .329 | 995.3 | 327.7 | .329 |
| - AB-MCTS-M$_{Full}$ | 2087.3 | 1651.3 | .791 | 4190.7 | 3411.7 | .814 | 6481.3 | 5130.0 | .792 | 438.0 | 130.7 | .298 | 915.0 | 263.7 | .288 | 1392.7 | 394.0 | .283 |
| - AB-MCTS-M | 484.7 | 371.3 | .766 | 1267.3 | 1009.7 | .797 | 2193.0 | 1766.7 | .806 | 160.7 | 44.3 | .276 | 349.7 | 101.0 | .289 | 547.0 | 158.0 | .289 |
| - MCTS | 2165.0 | 1956.0 | .903 | 4691.3 | 4143.0 | .883 | 7267.3 | 6321.3 | .869 | 308.3 | 131.3 | .426 | 834.3 | 361.3 | .433 | 1385.0 | 557.3 | .400 |
| - LiteSearch-Incre.[†] | 1174.0 | 356.3 | .304 | 1467.0 | 438.3 | .299 | 1582.0 | 463.7 | .293 | **1277.0** | 69.3 | .054 | **1488.7** | 79.3 | .053 | 1563.3 | 88.7 | .057 |
| - LiteSearch-Batch[†] | 935.3 | 288.3 | .308 | 1276.7 | 339.7 | .266 | 1371.3 | 352.7 | .257 | 781.7 | 28.7 | .037 | 1012.7 | 37.3 | .037 | 1167.7 | 44.0 | .038 |
| - BG-MCTS (ours) | **3660.7** | **3420.3** | **.934** | **8438.0** | **7783.0** | **.922** | **12825.0** | **11781.3** | **.919** | 438.3 | **236.3** | **.539** | 1187.0 | **658.3** | **.555** | **2012.3** | **1148.7** | **.571** |

# M. Qualitative Tree Visualizations

## M.1. Qualitative Comparison of MCTS and BG-MCTS

Figures 15 and 16 visualize example search trees produced by MCTS and BG-MCTS under the same fixed budget of $B = 20K$, using Llama-3.1-8B-Instruct on MATH500 Level-5 tasks. Within each figure, both methods are applied to the same problem instance. Figure 16 corresponds to the detailed tree visualization discussed in Section 5 (Fig. 7).

As shown in Figures 15 and 16, standard MCTS does not condition its expansion pattern on the remaining budget. Consequently, it continues to expand shallow nodes even late in the search, leaving insufficient budget to deepen promising branches and reach completed answers. In contrast, BG-MCTS explores broadly in the early stages and then shifts toward deeper expansion within selected promising subtrees as the budget is consumed. These examples illustrate the intended budget-aligned wide-to-deep behavior of BG-MCTS.

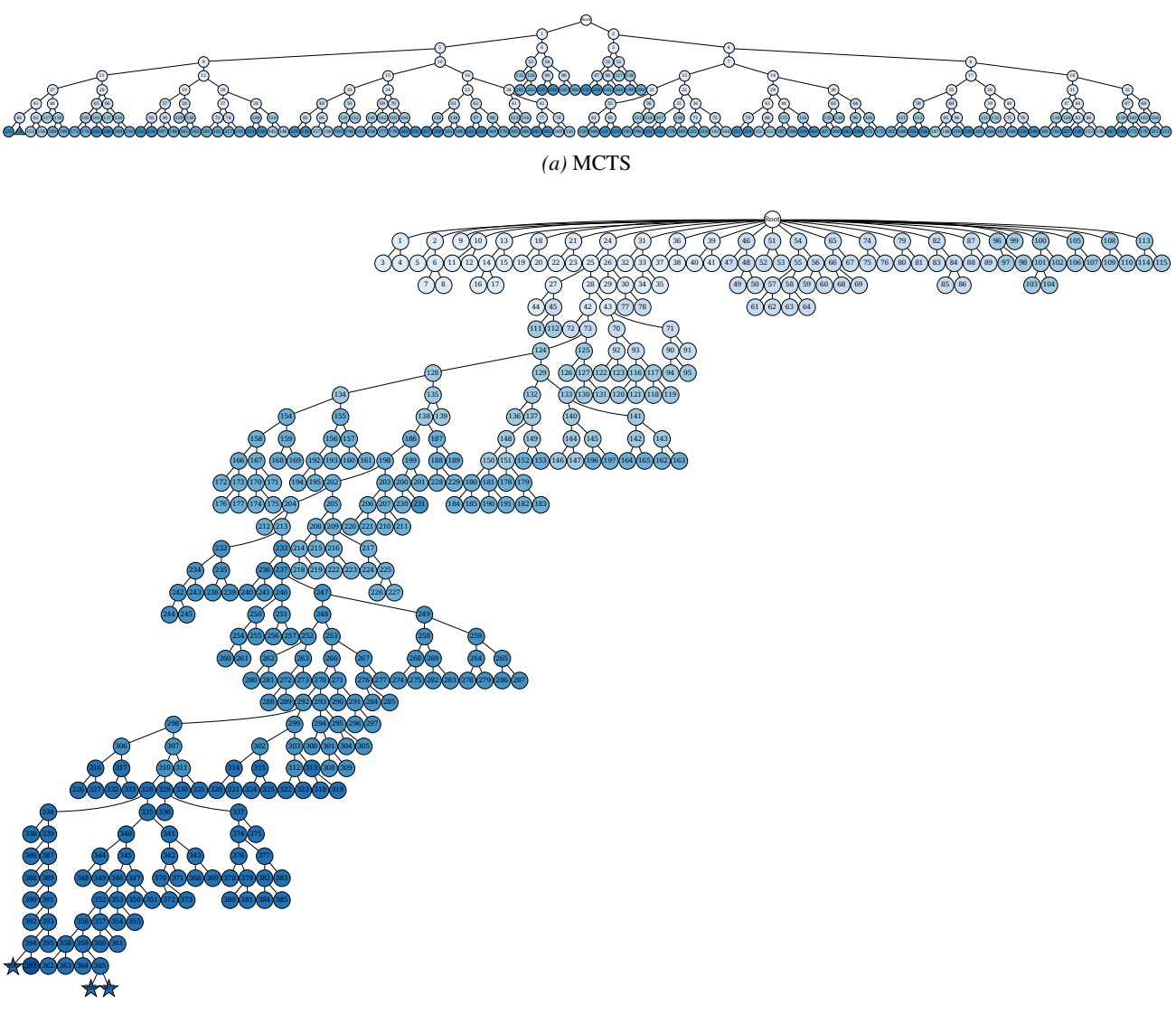

*(a)* MCTS

*(b)* BG-MCTS

*Figure 15.* **Example search trees for MCTS and BG-MCTS.** Both methods are run on the same MATH500 Level-5 problem using Llama-3.1-8B-Instruct under a fixed budget of $B = 20K$. Stars and triangles indicate correct and incorrect final-answer nodes, respectively. Darker nodes are expanded later in the search, when less budget remains; node labels indicate expansion order.

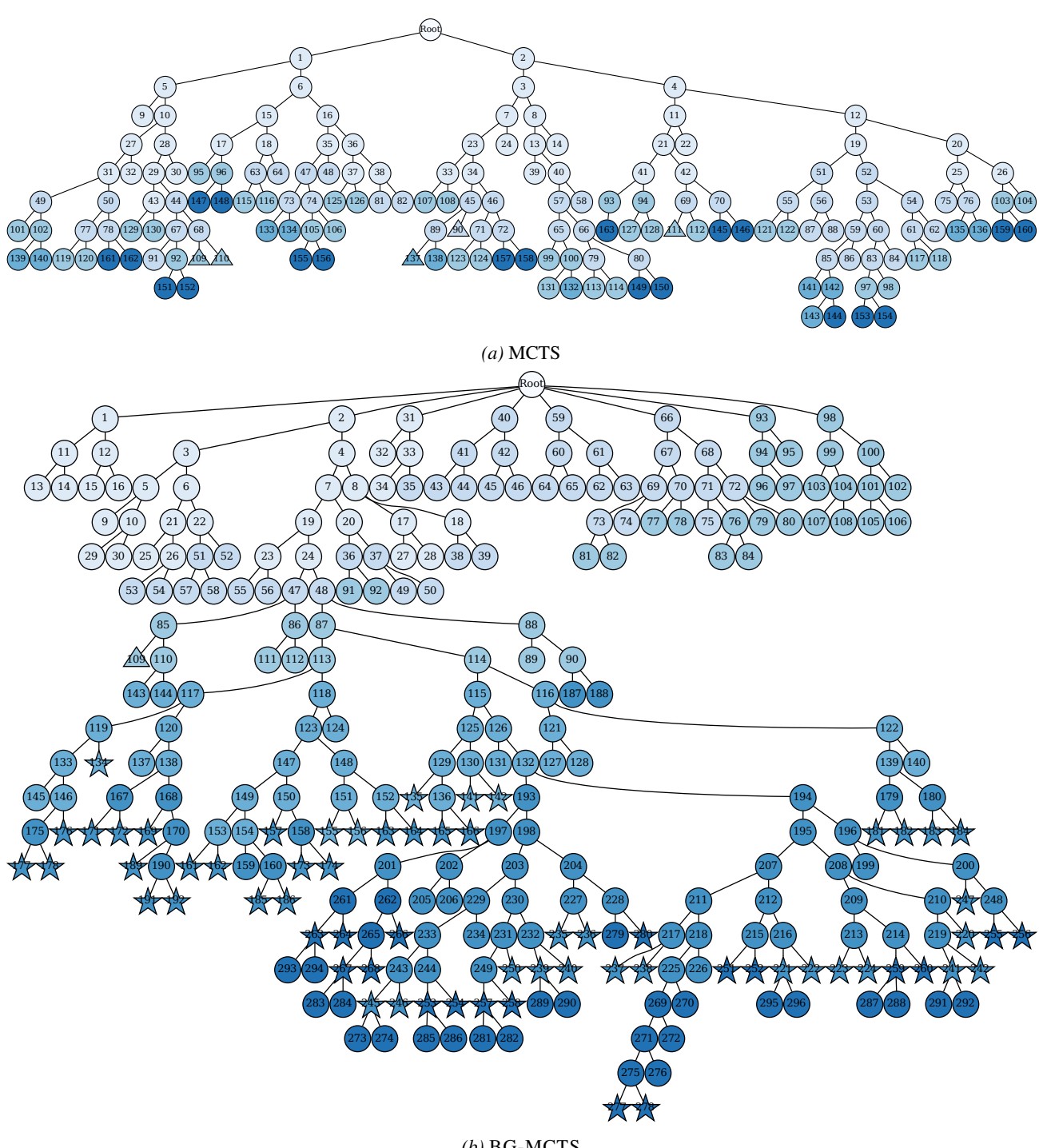

*(a)* MCTS

*(b)* BG-MCTS

*Figure 16.* **Example search trees for MCTS and BG-MCTS.** Both methods are run on the same MATH500 Level-5 problem using Llama-3.1-8B-Instruct with a fixed budget of $B = 20K$. Stars and triangles indicate correct and incorrect final-answer nodes, respectively. Darker nodes are expanded later in the search, when less budget remains; node labels indicate expansion order.

## M.2. Qualitative Analysis of Baseline Algorithms

Figures 17 and 18 visualize example search trees for Repeated Sampling, AB-MCTS-M, and LiteSearch under the same fixed budget of $B = 20K$, using Llama-3.1-8B-Instruct on MATH500 Level-5 tasks. Within each figure, all methods are applied to the same problem instance. Figures 15 and 17 show results on the same problem instance, and Figures 16 and 18 show results on another shared problem instance.

Repeated Sampling generates independent full trajectories, providing broad coverage over complete solutions but without reusing intermediate reasoning states or adapting expansion based on partial progress. AB-MCTS-M, by contrast, exhibits a strong breadth-oriented pattern: it repeatedly expands shallow nodes and performs relatively little deepening. This behavior is consistent with the low answer-reach rate observed in Figs. 4 and 11, and the visualizations show that AB-MCTS-M continues introducing shallow branches even late in the budget.

LiteSearch shows a different pattern. It tends to deepen early, reaching completed answers quickly, and only later broadens the search after reaching sufficient depth. This design is effective for reducing token usage and enabling early stopping, but it can also limit the exploration of diverse reasoning paths and lead to premature commitment. This depth-oriented behavior is consistent with the answer-reach trends observed in Figs. 4 and 11.

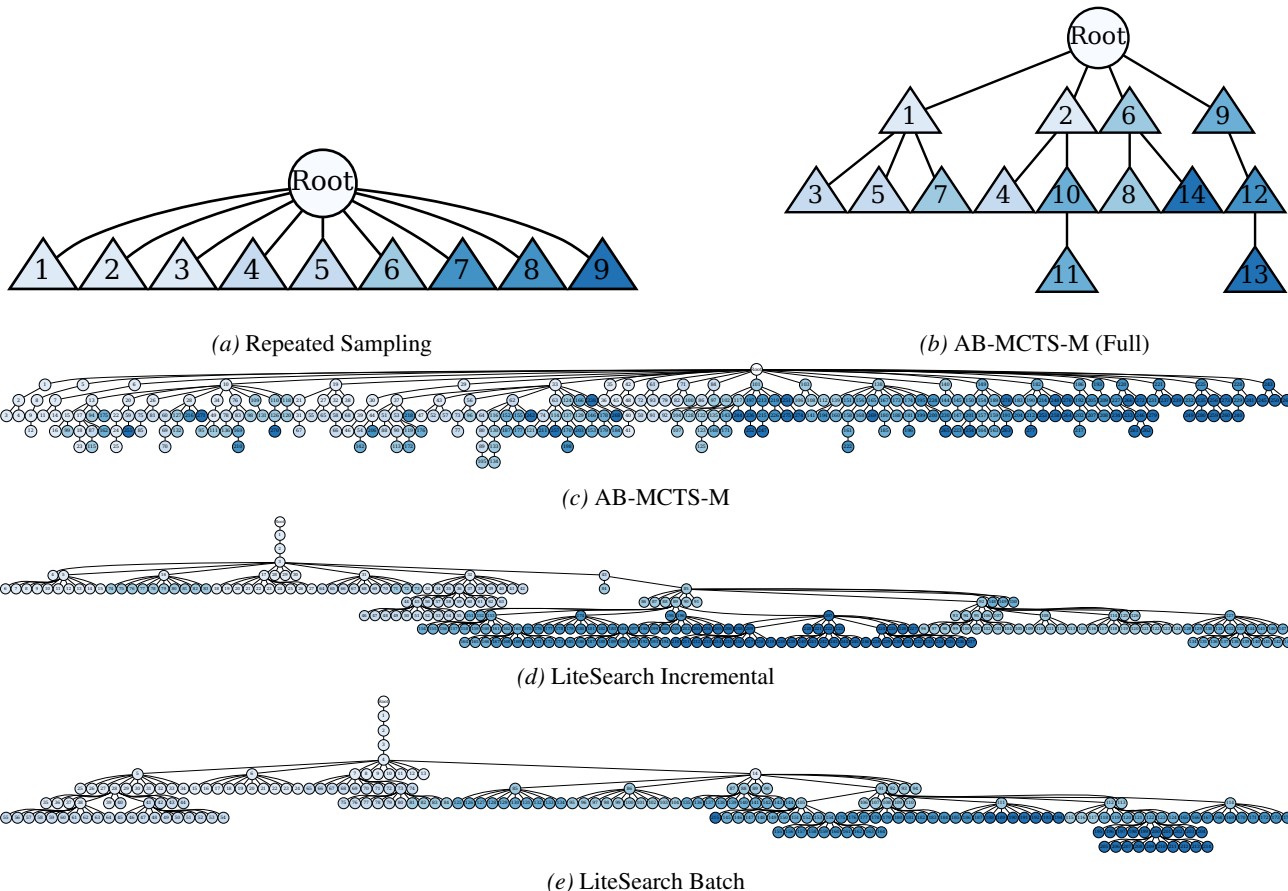

*(a)* Repeated Sampling

*(b)* AB-MCTS-M (Full)

*(c)* AB-MCTS-M

*(d)* LiteSearch Incremental

*(e)* LiteSearch Batch

*Figure 17.* **Example search trees for Repeated Sampling, AB-MCTS-M, and LiteSearch.** All methods are run on the same MATH500 Level-5 problem using Llama-3.1-8B-Instruct with a fixed budget of $B = 20K$. Stars and triangles indicate correct and incorrect final-answer nodes, respectively. Darker nodes are expanded later in the search, when less budget remains; node labels indicate expansion order.

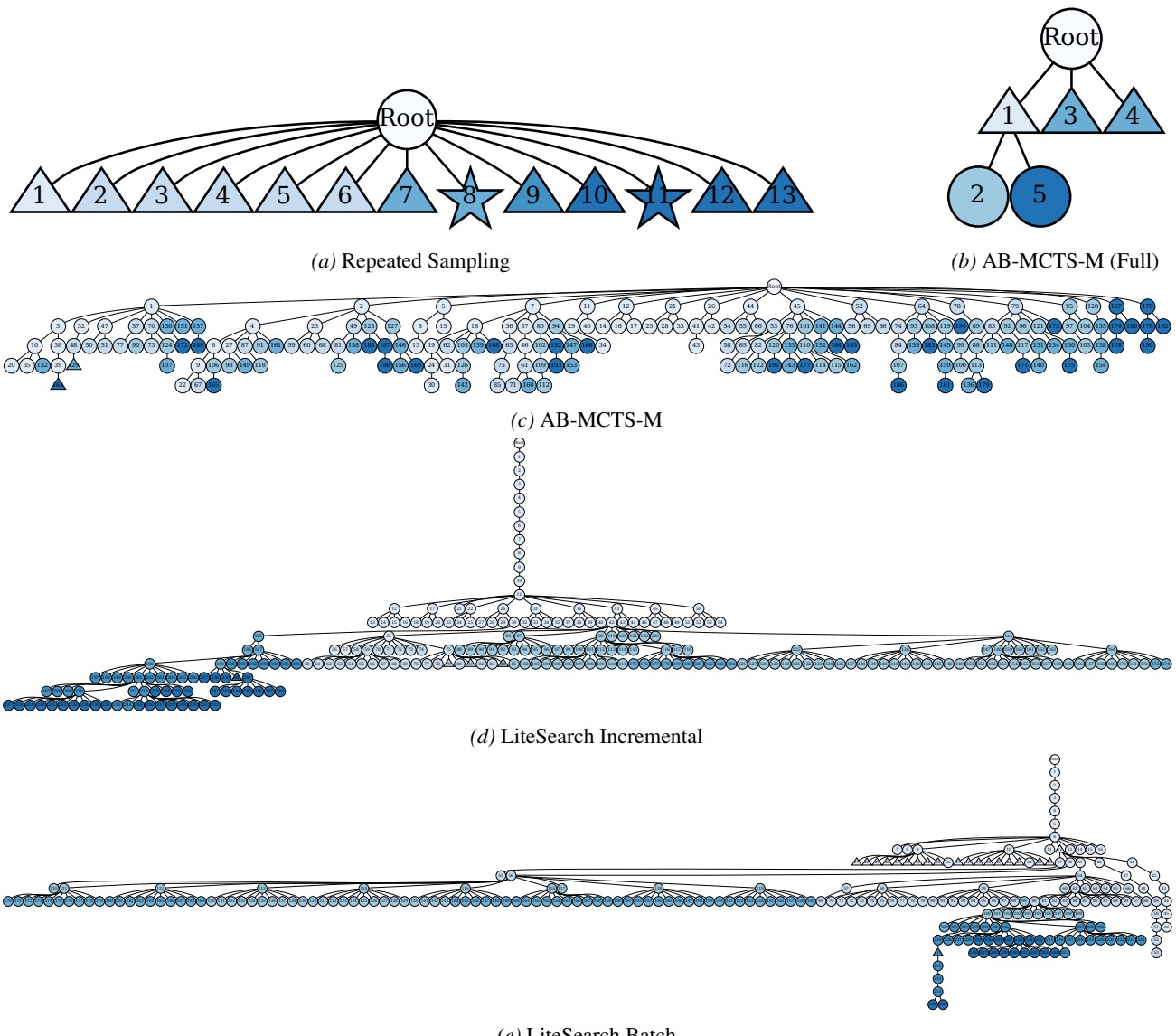

*(a)* Repeated Sampling

*(b)* AB-MCTS-M (Full)

*(c)* AB-MCTS-M

*(d)* LiteSearch Incremental

*(e)* LiteSearch Batch

*Figure 18.* **Example search trees for Repeated Sampling, AB-MCTS-M, and LiteSearch.** All methods are run on the same MATH500 Level-5 problem using Llama-3.1-8B-Instruct with a fixed budget of $B = 20K$. Stars and triangles indicate correct and incorrect final-answer nodes, respectively. Darker nodes are expanded later in the search, when less budget remains; node labels indicate expansion order.

