# OpenReview forum: "Aligning Tree-Search Policies with Fixed Token Budgets in Test-Time Scaling of LLMs"
_ICML.cc/2026/Conference — ICML 2026 regular_

### Official Review · Reviewer_pWat · 2026-02-15

**Soundness:** 3
**Presentation:** 3
**Significance:** 3
**Originality:** 2
**Overall Recommendation:** 4
**Confidence:** 3

**Summary:**

This paper addresses the problem of tree-search–based test-time scaling for LLMs under fixed per-query token budgets. Existing MCTS-style decoding methods are largely budget-agnostic, using the token budget only as a stopping condition, which can lead to inefficient late-stage branching or insufficient refinement of promising solutions. The key challenge is to allocate limited inference tokens effectively across exploration and completion. To address this, the authors propose Budget-Guided MCTS (BG-MCTS), which conditions both node selection and tree widening on the remaining budget, gradually shifting from broad exploration to depth-focused refinement as the budget depletes. Experiments on mathematical reasoning benchmarks show that BG-MCTS consistently outperforms budget-agnostic tree-search baselines under fixed token constraints.

**Compliance With Llm Reviewing Policy:**

Affirmed.

**Final Justification:**

Following the rebuttal I believe the paper should be accepted, but I don't have a strong opinion

**Key Questions For Authors:**

**Questions**
* **Q.1** Have you evaluated BG-MCTS on domains beyond mathematical reasoning (coding or general instruction-following)? If so, do the improvements persist? Evidence of consistent gains in other domains would strengthen the paper.
* **Q.2.** Do you have results on more recent and stronger models (larger open-weight models or recent frontier APIs)? If BG-MCTS continues to provide gains in stronger-model regimes, this would increase confidence that the method scales and remains relevant.
* **Q.3.** Did you compare against simpler heuristics, such as manually scheduling a fixed exploration coefficient over time without modifying widening?
* **Q.4.** How sensitive is BG-MCTS to the added hyperparameters?

**Strengths And Weaknesses:**

**Strengths**
* **S.1** The proposed Budget-Guided are clearly defined, coherent, and well integrated. The ablations and controlled fixed-budget comparisons support the main claims.
* **S.2.** The paper compares against strong and relevant baselines under consistent token budgets.
* **S.3.** Conditioning search behavior on remaining compute is a useful idea that could be adopted or extended in other inference-time scaling methods.

**Weaknesses**
* **W.1.** Experiments are restricted to math reasoning benchmarks. It remains unclear whether the approach generalizes to other domains such as coding or broader instruction-following tasks.
* **W.2.** The evaluation relies on relatively small and somewhat outdated models, which weakens the empirical impact. Results on newer, stronger models (such as Qwen 3) would strengthen the paper.
* **W.3.** The core idea of annealing exploration and widening based on remaining budget is intuitive and heuristic. While well executed, the contribution feels incremental rather than conceptually novel.
* **W.4.** The work is primarily empirical and lacks theoretical grounding or deeper analysis of why the proposed scheduling is optimal.

---

> ### Author Rebuttal · Authors · 2026-03-31
>
> We thank the reviewer for the constructive feedback and concrete suggestions. Below we clarify the scope of the method and provide additional evidence on a stronger recent model and on hyperparameter sensitivity.
>
> ## **W1/Q1: Scope beyond mathematical reasoning**
> We agree that the current empirical scope is limited to math.
> We have not yet completed non-math evaluations within the rebuttal period.
> However, in formulation, BG-MCTS itself is domain-agnostic:
> Algorithm~1 only assumes a scalar evaluation signal $Q(x)$ for partial states.
> Thus, the main bottleneck for domains such as coding or broader instruction-following is not the budget-conditioned search policy itself, but the availability of reliable intermediate evaluation (e.g., tests, verifiers, execution feedback, or task-specific reward models).
> In the camera ready version, we will revise the paper to make this scope explicit and soften any over-general claim.
>
> ## **W2/Q2: Results on a stronger recent model**
> Our original experiments intentionally focused on open-weight sub-10B models (e.g., LLama-3.1-8B-Instruct),
> *where scaling model size is often impractical and test-time scaling is the more realistic lever.*
> In this response, we additionally evaluated a stronger recent model, Qwen3-8B [An Yang et al., 2025], on MATH500 (see the table below).
> BG-MCTS remains best at 10K and 20K, and remains competitive at 30K:
>
> | Qwen3-8B | 10k | 20k | 30k |
> |---|---:|---:|---:|
> | Greedy | .649 | .649 | .649 |
> | AB-MCTS-M\_Full | .669 | .689 | **.707** |
> | AB-MCTS-M | .403 | .460 | .500 |
> | MCTS | .624 | .664 | .697 |
> | BG-MCTS | **.672** | **.704** | .697 |
>
> > **Table:** These results were obtained by solving Math500 Level 5 with Qwen3-8B. Each algorithm was evaluated over three runs, and the reported values are the averages across those runs.
>
> These results suggest that the benefit of budget guidance is not confined to weaker models; it remains useful in a stronger-model regime.
>
> ## **W3/Q3: Novelty and comparison to a simpler heuristic**
> We agree that the core intuition is simple.
> Our claim is not that ``annealing exploration as compute runs out'' is abstractly new,
> but that fixed-budget LLM tree decoding benefits from treating the remaining budget ratio $\rho$ as a first-class control variable.
> BG-MCTS conditions both selection and widening on $\rho$, instead of using the budget only as a stopping condition (Eqa.3--7).
>
> Question-3 is also partly addressed by our existing ablation (Table 2): using only exploration annealing (Eq.~3) already helps, but it does not match the full BG-MCTS.
> This indicates that a simple time-varying exploration coefficient alone is insufficient; completion shaping and budget-guided widening are complementary.
>
> To directly answer Q3, we additionally tested a simpler variant that keeps budget-aware selection but removes the generative-child term (Eq.~6), i.e., it anneals exploration over time without modifying widening.
> As shown in the Table below, this variant is overall weaker and less robust than full BG-MCTS, indicating that exploration annealing alone does not explain the gain.
>
> | methods | 10k | 20k | 30k |
> |---|---:|---:|---:|
> | mcts | .619 | .657 | .659 |
> | BG-MCTS w/o widening | .679 | .619 | (running) |
> | BG-MCTS | .662 | .699 | .711 |
>
> > **Table**: These are the experimental results obtained using Qwen2-7B-Instruct as the generation model on MATH500 Level 5. The results for MCTS and BG-MCTS are taken directly from the paper, while the result for BG-MCTS w/o widening is based on a single experimental run.
>
> ## **W4/Q4: Theory and sensitivity**
> We not claim optimality or regret guarantees. The contribution of this paper is an empirically validated decoding policy for fixed-budget search, motivated by two monotonic design principles:
> as the remaining budget decreases, late-stage branching should be suppressed, and deeper promising branches should receive increasing preference.
> We will clarify this positioning in the camera ready version.
>
> Regarding sensitivity, BG-MCTS is not brittle in a moderate neighborhood of the default setting.
> In the main paper, we keep $c=\sqrt{2}$ and $\kappa=\lambda=1$ fixed across experiments.
> In an additional sweep on Qwen-2.5-7B / MATH500 with $B=20$K, accuracy ranges from 0.664 to 0.709 across the tested settings, with several nearby configurations matching or exceeding the default.
>
> | $c$ | $\kappa$ | $\lambda$ | Score |
> |---:|---:|---:|---:|
> | $\sqrt{2}$ | 1 | 1 | .699 |
> | $\sqrt{2}$ | 1 | 0.5 | .694 |
> | $\sqrt{2}$ | 1 | 1.5 | .694 |
> | $\sqrt{2}$ | 0.5 | 1 | .709 |
> | $\sqrt{2}$ | 0.5 | 0.5 | .664 |
> | $\sqrt{2}$ | 0.5 | 1.5 | .664 |
> | 1 | 1 | 1 | .664 |
> | 1 | 1 | 0.5 | .694 |
> | 1 | 1 | 1.5 | .687 |
> | 1 | 0.5 | 1 | .694 |
> | 1 | 0.5 | 0.5 | .694 |
> | 1 | 0.5 | 1.5 | .687 |
>
> > **Table**: Sensitivity of BG-MCTS to hyperparameters under fixed budget (B=20K) with Qwen-2.5-7B on MATH500 level.5. the result is based on a single experimental run.

---

> > ### Author Rebuttal · Reviewer_pWat · 2026-03-31
> >
> > Thank you for the additional results and explanations. These satisfy most of my concerns. I do think the novelty here is still weak, but the additional experiments clearly show that simple heuristics are not sufficient. I appreciate the rebuttal efforts and therefore will be updating my score. I believe that following these results the paper should be accepted.

---

> > > ### Author Response · Authors · 2026-04-02
> > >
> > > Thank you very much for the careful follow-up and for updating your assessment. We appreciate your candid feedback on the novelty, and we are glad that the additional experiments clarified that the gains are not explained by simple heuristics alone. We will sharpen the positioning of our contribution in the final version. Thank you again for your thoughtful evaluation and support.

---

### Official Review · Reviewer_7zyP · 2026-03-04

**Soundness:** 3
**Presentation:** 4
**Significance:** 4
**Originality:** 3
**Overall Recommendation:** 4
**Confidence:** 3

**Summary:**

This paper proposes Budget-Guided MCTS (BG-MCTS), a tree-search decoding method designed to optimize Large Language Model (LLM) reasoning under fixed token budgets. BG-MCTS dynamically adjusts its search policy based on the remaining budget. By introducing a "budget sufficiency ratio" to anneal exploration and a "virtual generative child" to control tree widening, the algorithm balances the trade-off between search breadth and depth. This strategy ensures that the model explores diverse reasoning paths early on and prioritizes completion as the budget depletes, achieving consistently strong performance under various fixed budget constraints.

**Compliance With Llm Reviewing Policy:**

Affirmed.

**Final Justification:**

I recommend a Weak Accept for this paper. Overall, the submission makes a practical contribution to LLM test-time scaling by proposing an budget-aware tree-search decoding method (BG-MCTS). The paper is good in presentation and significance, effectively addressing a critical real-world deployment constraint with intuitive mechanisms.

However, my assessment of its soundness and originality remains moderate. The empirical validation is strictly confined to mathematical reasoning. It did not provide cross-domain experiments or rigorous PRM stress-testing.

In conclusion, this is a good paper for its specific domain, but the limited empirical scope prevents a higher rating.

**Key Questions For Authors:**

1.Does the calculation of the generative score (Eq. 6) introduce significant latency overhead compared to the inference time of the LLM?
2.The proposed BG-MCTS algorithm relies heavily on the scalar values from GenPRM-7B to guide selection and widening. How robust is the method to the quality of the Process Reward Model?
3.The empirical evaluation focuses exclusively on mathematical reasoning tasks (MATH500 and AIME24/25), where reasoning steps can be relatively structured. Does your strategy generalize to other complex domains, such as code generation or open-ended logical deduction?

**Limitations:**

Yes

**Strengths And Weaknesses:**

1.The paper targets a realistic scenario in test-time scaling: fixed token budgets. It optimizes the search behavior within a strict operational constraint, making the findings valuable for real-world LLM deployment.
2.The authors achieve adaptive search behavior smoothly by introducing the budget sufficiency ratio and a virtual generative node. This design seamlessly integrates into the MCTS framework without introducing complex settings.
3.The ablation study (Table 2) effectively isolates and validates the contribution of each proposed component. The detailed analysis of tree depth and width dynamics over time (e.g., Figures 4, 10, and 11) shows that the algorithm successfully executes the intended "wide-to-deep" exploration strategy.

---

> ### Author Rebuttal · Authors · 2026-03-31
>
> We thank the reviewer for the positive assessment and for highlighting the practical fixed-budget setting, the simplicity of the integration into MCTS, and the supporting ablation/analysis.
>
> ##  **Q1: Latency overhead of Eq.~6**
>
> No. The generative score in Eq.~6 introduces negligible overhead compared to LLM decoding and PRM evaluation.  It is computed from already available child statistics and requires no additional LLM or PRM forward pass. In practice, the dominant cost remains generation and reward evaluation.
>
> ## **Q2: Robustness to PRM quality**
>
> We agree that PRM quality is important.
>
> First of all, in all our experiments, standard MCTS and BG-MCTS use the same GenPRM-7B, so the gain of BG-MCTS comes from better budget-aware allocation under the same reward signal, rather than from a stronger evaluator.
>
> In this response, we inspected the PRM scores and found them to be highly saturated / near-binary (see the table below), making this a challenging regime.
> Therefore, our experimental results in the submission suggest that BG-MCTS remains beneficial even with an imperfect PRM, and stronger PRMs would likely further improve both methods.
>
> | task | p10 | p25 | p50 | p75 | p90 |
> |---|---:|---:|---:|---:|---:|
> | Math500 lv.5 | $2.183 \cdot 10^{-3}$ | $9.740 \cdot 10^{-1}$ | $9.996 \cdot 10^{-1}$ | $9.988 \cdot 10^{-1}$ | $9.996 \cdot 10^{-1}$ |
> | AIME24/25 | $0.130 \cdot 10^{-4}$ | $1.170 \cdot 10^{-3}$ | $9.399 \cdot 10^{-1}$ | $9.968 \cdot 10^{-1}$ | $9.990 \cdot 10^{-1}$ |
>
> > *Table:* The percentiles of GenPRM evaluation scores observed when solving Math500 Level 5 and AIME24/25 under a fixed 30k-token budget using Qwen2.5-7B-Instruct.
>
> ## **Q3: Generalization beyond math**
>
> Our current experiments are limited to mathematical reasoning, so **we do not claim broad empirical generalization in the present submission.** That said, **the BG-MCTS mechanism itself is not math-specific.** Algorithm 1 only assumes that each expanded partial state $s$ can be assigned a scalar evaluation $Q(s)$ through `EVALUATE(s)`; this evaluation can come from a verifier, reward model, execution feedback, or a domain heuristic. Eq. 3 uses this signal in budget-conditioned selection, Eq.~5 adds a late-stage depth bias, and Eq. 6 compares deepening vs. widening using only the mean and variance of current child scores under the same remaining-budget ratio $\rho$.
>
> Therefore, **whenever a task admits (i) incremental expansion into partial states and (ii) some intermediate evaluation signal, the same budget-guided control applies in principle**. For example, in code generation, $Q(s)$ can be derived from unit tests, compiler feedback, or execution outcomes; in logical deduction or multi-hop reasoning, it can come from a verifier or heuristic consistency signal. In this sense, the main bottleneck outside math is not the BG-MCTS policy itself, but obtaining reliable intermediate evaluation and task-appropriate step boundaries.

---

> > ### Author Rebuttal · Reviewer_7zyP · 2026-04-02
> >
> > Thank you for additional explanations. My concern regarding the latency overhead is addressed. However, the rebuttal still lacks empirical experiments in cross-domain tasks, and robustness to varying PRM qualities was argued but not tested empirically. I will remain my score.

---

> > > ### Author Response · Authors · 2026-04-07
> > >
> > > Thank you for the follow-up.
> > >
> > > We agree that the current submission does not empirically establish cross-domain generality or robustness across multiple PRM qualities, and we will make these limitations more explicit in the final version. Our point in Rebuttal Q3 was only that BG-MCTS is not specific to math in its formulation: it requires sequential generation and an evaluator for partial outputs, which is the same basic applicability condition assumed by search baselines such as MCTS and AB-MCTS. Thus, the remaining limitation is empirical scope rather than task-specific dependence of the method itself.
> > >
> > > For PRM robustness, our intent was not to claim robustness across evaluators, but to show that the current GenPRM already exhibits saturated, weakly discriminative score distributions. The fact that BG-MCTS still improves over standard MCTS under this same challenging evaluator suggests that its benefit comes from budget-aware search, not from a stronger reward signal. We agree, however, that experiments with multiple evaluators would strengthen this claim, and we will clarify this limitation in the final version.

---

### Official Review · Reviewer_3KCj · 2026-03-13

**Soundness:** 3
**Presentation:** 3
**Significance:** 2
**Originality:** 2
**Overall Recommendation:** 3
**Confidence:** 4

**Summary:**

The authors propose Budget-Guided MCTS (BG-MCTS), a tree-search decoding algorithm that explicitly conditions its node-selection policy on the remaining token budget. The key idea is to dynamically shift the search behavior over time: When the budget is large, the algorithm encourages broader exploration, selecting shallower nodes and expanding multiple candidate continuations; When the budget becomes small, it prioritizes deeper nodes and refinement, aiming to complete promising reasoning trajectories rather than continuing to branch.

This adaptive policy aims to avoid late-stage over-branching (i.e., the search continues expanding new candidates even though there is not enough budget to complete them) and premature termination (i.e., promising reasoning chains are abandoned due to misallocated exploration).

The method is evaluated on mathematical reasoning benchmarks (MATH500 and AIME24/25) using open-weight LLMs. The authors report consistent improvements over budget-agnostic tree-search baselines across different token budgets.

**Compliance With Llm Reviewing Policy:**

Affirmed.

**Key Questions For Authors:**

How does the algorithm compare with fixed-depth tree search and dynamic beam search?

How does the approach help with other tasks like coding, physics or other tasks?

**Limitations:**

Partially. I also include some additional limitations above.

**Strengths And Weaknesses:**

Strengths:
1. Addresses a realistic deployment constraint: A major strength is the focus on fixed token budgets, which reflects real-world inference settings where compute is limited. Much of the literature assumes unlimited or loosely bounded compute, so framing the search policy around budget constraints is both practical and timely.
2. Conceptually clean idea: The central insight—that search behavior should evolve as the remaining compute decreases—is intuitive and well motivated. The transition from exploration → refinement mirrors strategies used in classical planning and MCTS literature.
3. Simple modification with clear implementation: BG-MCTS modifies node selection rather than redesigning the entire decoding pipeline. This makes the method easy to integrate with existing tree-search decoders, compatible with multiple LLM architectures.

Weaknesses:
1. Limited novelty relative to classical MCTS ideas: While the paper frames the approach in the context of LLM inference, the core idea—budget-aware search scheduling—is conceptually similar to resource-aware strategies studied in classical search and planning. The paper could more clearly position itself relative to that literature.
2. Heuristic design lacks strong theoretical justification: The budget-dependent node selection policy appears largely heuristic. The paper does not provide theoretical guarantees, regret analysis, or formal justification for the specific scheduling function. As a result, it is unclear whether the proposed schedule is optimal or merely empirically effective.
3. Evaluation scope is narrow: The experiments focus primarily on math reasoning benchmarks. It remains unclear whether the method generalizes to coding tasks, open-ended reasoning, planning tasks, multi-modal reasoning.
4. Stronger baseline: While the paper compares against budget-agnostic tree search, it would be valuable to include comparisons with compute-adaptive decoding strategies, other test-time scaling methods (e.g., self-consistency or best-of-N sampling), or hybrid search approaches. Without these comparisons, the empirical advantage is somewhat harder to contextualize.

---

> ### Author Rebuttal · Authors · 2026-03-31
>
> We thank the reviewer for the thoughtful feedback and for recognizing the practical relevance of the fixed-budget setting and the simplicity of our modification.
> We have carefully addressed all the weaknesses and questions raised.
>
> ## **W1: Novelty relative to classical MCTS/search**
>
> We agree that resource-aware search has important roots in the classical MCTS/search literature, and we do not claim that the *abstract idea* of adapting search to remaining compute is itself new. Our contribution is narrower: we instantiate this idea for LLM test-time scaling under a fixed output-token budget, where search operates over partial reasoning traces and must decide both whether to deepen an existing branch and whether to widen the tree with a new continuation. Concretely, BG-MCTS conditions both selection and widening on the remaining-budget ratio $\rho$ (Eqs. 3--7), rather than using budget only as a stopping rule as LiteSearch [Wang et al., 2024a]. We will revise the paper to position this connection more explicitly and earlier.
>
>
> ## **W2: Heuristic design**
>
> We do not claim optimality or regret guarantees. Our goal is a simple budget-consistent policy for fixed-budget decoding.   The design follows two monotonic principles: as $\rho$ decreases, exploration should be annealed and late-stage widening should be suppressed, while deeper promising nodes should receive increasing preference.  Thus BG-PUCT behaves close to standard PUCT when budget is ample and shifts smoothly toward refinement as budget runs down. A full theory for LLM tree search with learned PRMs, stochastic token costs, and dynamic widening is beyond the current paper; we will clarify that our contribution is an empirically validated decoding policy rather than a theoretically optimal schedule.
>
> ## **W3/Q2: Applicability beyond math**
>
> We agree that the current empirical scope is limited to math. However, BG-MCTS itself is domain-agnostic: Algorithm 1 only requires an evaluation signal $Q(x)$ for partial states, which can come from a verifier, reward model, execution feedback, or a domain heuristic. Therefore the same budget-conditioned control can in principle apply to coding (e.g., tests/compilers), physics (e.g., symbolic/tool-based checking), and planning tasks. The main bottleneck is reliable intermediate evaluation, not the search policy itself. We will clarify this scope and soften generality claims accordingly.
>
> ## **W4/Q1: Comparison to fixed-depth and beam-style search**
>
> To answer Q1, we compare BG-MCTS against two simpler search families under the same fixed-budget protocol.
>
> First, for *rollout-only fixed-depth search*, the relevant baseline is Repeated Sampling / Best-of-$N$, which is already included in our submission (Sec. 4.1, Table 1). In this setting, each rollout independently generates a complete solution using Full generation, without intermediate branching or node reuse. BG-MCTS consistently outperforms this fixed-depth baseline under the same budget.
>
> Second, we additionally evaluated a *beam-style* baseline in the rebuttal: standard beam search under the same prompt/model/budget protocol. The result below shows that it is also weaker than BG-MCTS.
>
> | Method | 10K | 20K | 30K |
> |---|---:|---:|---:|
> | Repeated Sampling (fixed-depth rollout-only) | .632 | .664 | .702 |
> | Beam Search | .328 | .537 | .597 |
> | BG-MCTS | **.662** | **.699** | **.711** |
>
> > *Table:* Additional baseline comparison on Qwen-2.5-7B-Instruct / MATH500 (Lv.5) under the same fixed-budget protocol as Table 1. We conducted the Beam Search experiments with a beam width of 5 and a branching factor of 10. The results for BG-MCTS and Repeated Sampling are identical to those reported in the paper, whereas the Beam Search result is based on a single experimental run.
>
> This result is consistent with the design of BG-MCTS: unlike repeated sampling, it reuses intermediate reasoning states; and unlike beam-style frontier pruning, it can revisit arbitrary internal nodes and explicitly decide whether to deepen an existing branch or widen the tree as the remaining budget decreases.

---

### Official Review · Reviewer_J8Nq · 2026-03-13

**Soundness:** 3
**Presentation:** 3
**Significance:** 3
**Originality:** 3
**Overall Recommendation:** 4
**Confidence:** 3

**Summary:**

This paper proposes Budget-Guided MCTS (BG-MCTS), a tree-search decoding algorithm that aligns search behavior with fixed per-query token budgets. Unlike standard MCTS approaches that treat the budget only as a termination condition, BG-MCTS conditions both node selection and tree widening on the remaining budget ratio $\rho = 1 - C_{\text{used}}/B$.

The method introduces three key mechanisms: (1) Budget-annealed exploration in the PUCT score, where the exploration bonus decays as $\rho \to 0$; (2) Completion bias that increasingly prioritizes deeper nodes as the budget depletes; and (3) Budget-guided widening that suppresses generating new branches late in the search. Together, these induce a "wide-to-deep" schedule: broad exploration early, followed by focused refinement and answer completion.

Empirically, BG-MCTS is evaluated on MATH500 (Level 5) and AIME24/25 using Llama-3.1-8B-Instruct and Qwen-2.5-7B-Instruct under fixed budgets of $B \in \{10\text{K}, 20\text{K}, 30\text{K}\}$ tokens. Across 12 settings, BG-MCTS achieves best or second-best accuracy in 11 cases, outperforming budget-agnostic baselines (standard MCTS, AB-MCTS-M, LiteSearch) and remaining competitive with sampling-based methods. Ablations confirm all three components are necessary, and analysis shows BG-MCTS shifts from breadth to depth as budget depletes, achieving higher precision in answer generation.

**Compliance With Llm Reviewing Policy:**

Affirmed.

**Key Questions For Authors:**

## Key Questions For Authors

1. **Hyperparameter Sensitivity:** The paper sets $\kappa = \lambda = 1$ for the completion bias (Eq. 5) and widening score (Eq. 6). How sensitive is performance to these values across different budget scales (e.g., very small budgets like $B = 1\text{K}-5\text{K}$ tokens or much larger ones)? Do you expect these hyperparameters to transfer across domains without tuning?

2. **Generalization Beyond Math:** Have you evaluated BG-MCTS on non-mathematical reasoning tasks (e.g., code generation, commonsense reasoning, or multi-hop QA)? The current reliance on GenPRM-7B (trained for math) limits the demonstration of general applicability. If not, what do you anticipate would be the main challenges in domains where intermediate step verification is less reliable?

3. **Input Token Budgets:** The current formulation focuses on output token budgets $B$. Many real-world deployments also face input context length constraints. How would BG-MCTS handle cases where the input prompt itself consumes a large portion of the available context window, effectively reducing the searchable tree depth?

4. **Dynamic Budget Allocation:** The paper assumes a fixed per-instance budget $B$. Have you considered scenarios where one could dynamically allocate budget across multiple instances (e.g., spending more tokens on harder problems and less on easier ones)? Would the budget-ratio conditioning $\rho$ require modification for such settings?

5. **Comparison with Early Stopping:** LiteSearch uses early stopping when RM scores exceed $\epsilon \geq 0.9$. In Table 1, you disable greedy pre-generation in LiteSearch to align token counting. How does BG-MCTS compare to LiteSearch with early stopping enabled, and would it make sense to combine budget-guidance with early stopping?

**Limitations:**

## Limitations

The authors adequately discuss limitations in Section 6, including the sparse reward landscape from GenPRM-7B, the focus on output tokens only, and the single-instance budget constraint.

One additional limitation to consider: The method assumes the availability of a reliable process reward model (PRM) for computing $Q(x)$. In domains where such verifiers are unavailable or expensive to run, the value estimates may be noisy, potentially affecting the budget-conditioned decisions.

Additionally, while the paper shows that BG-MCTS achieves higher precision in generated answers (Table 3), it sometimes reaches fewer answer nodes overall. This trade-off should be discussed more explicitly in the context of applications requiring high coverage versus high precision.

The paper could also discuss computational overhead—the budget tracking and modified score calculations are lightweight $O(1)$ per node, but the repeated evaluation of the PRM for each node expansion remains a significant cost factor.

**Strengths And Weaknesses:**

## Strengths

- **Technically Sound Methodology.** The proposed BG-MCTS introduces well-defined mathematical modifications to standard PUCT. The BG-PUCT score coherently integrates the budget sufficiency ratio $\rho = 1 - C_{\text{used}}/B$ into both exploration and exploitation terms. The completion bias and the virtual widening score are principled extensions that jointly induce the desired "wide-to-deep" behavior.

- **Comprehensive Empirical Validation.** The experiments cover 12 distinct settings (2 models $\times$ 2 benchmarks $\times$ 3 budget levels $B \in \{10\text{K}, 20\text{K}, 30\text{K}\}$). The ablation study (Table 2) rigorously demonstrates that all three components—exploration annealing (Eq. 3), completion shaping (Eq. 5), and widening annealing (Eq. 6)—are necessary, as removing any single component degrades performance.

- **Insightful Behavioral Analysis.** The paper provides convincing empirical evidence that BG-MCTS behaves as intended. Figures 4, 10, and 11 show that tree depth increases sharply while width growth decelerates as $\rho \to 0$, confirming the budget-aligned wide-to-deep transition. Table 3 demonstrates that BG-MCTS achieves significantly higher precision (correct-to-answered ratio) compared to baselines, reaching up to $51.8\%$ on AIME24/25 with $B=10\text{K}$ versus $11.3\%$ for standard MCTS.

- **Practical Relevance.** The work addresses a concrete deployment constraint—fixed per-query token budgets—that is often overlooked in tree-search literature. The "think broadly, finish strong" paradigm is particularly valuable for API-served models (e.g., GPT-4.5/5) where cost predictability is critical. The implications for high-quality data synthesis (Section 6) further extend the utility beyond immediate inference.

## Weaknesses

- **Limited Theoretical Analysis.** While the empirical results are strong, the paper lacks theoretical guarantees regarding the proposed algorithm. Specifically, there is no analysis of convergence properties, regret bounds under budget constraints, or optimality conditions for the budget-conditioned policy. A theoretical characterization of how $\rho$ affects the exploration-exploitation trade-off would strengthen the contribution.

- **Narrow Domain Evaluation.** The evaluation is restricted to mathematical reasoning (MATH500 and AIME24/25). It remains unclear how BG-MCTS would perform in other domains such as code generation, commonsense reasoning, or open-ended text generation, where intermediate step verification (required for computing $Q(x)$) may be less reliable or unavailable. The reliance on GenPRM-7B, which outputs near-binary scores as noted in Section 6, may limit generalizability.

- **Late Positioning Against Related Work.** The distinction from BATS (Liu et al., 2025b)—which also uses budget-aware prompting but for tool-augmented agents rather than tree-search decoding—is buried in Section 7 (Related Work). This comparison should appear earlier (e.g., in the Introduction or Methodology) to clarify the unique contribution of BG-MCTS relative to concurrent budget-aware approaches.

- **Sensitivity to Reward Model Quality.** The method assumes access to a reliable process reward model (PRM) for node evaluation. As acknowledged in Section 6, GenPRM-7B often produces sparse, near-binary scores that fail to distinguish among strong candidates. In such regimes, the budget-conditioned selection may be suboptimal because the value estimates $Q(x)$ lack discriminative power. The paper does not quantify how PRM quality affects BG-MCTS performance compared to standard MCTS.

- **Fixed Budget Assumption.** The current formulation assumes a fixed per-instance budget $B$ determined upfront. The paper does not address scenarios where the budget might be dynamically allocated across instances (e.g., spending more tokens on harder problems) or where the input prompt itself consumes a significant portion of the context window, effectively constraining the searchable tree depth.

---

> ### Author Rebuttal · Authors · 2026-03-31
>
> ## **W1: Limited Theoretical Analysis**
> We agree that the current paper does not provide formal convergence, regret, or optimality guarantees. Our claim is not that the proposed schedule is theoretically optimal, but that it is a simple and principled decoding policy for fixed-budget search: as the remaining budget decreases, exploration and late-stage widening should be suppressed, while deeper promising branches should receive increasing preference. We will clarify this positioning more explicitly in the revision.
>
> ## **W2/Q2: Scope beyond mathematical reasoning**
> BG-MCTS is not math-specific: it only requires a scalar evaluation $\(Q(s)\)$ for partial states. Thus, it can in principle apply to any task with incremental expansion and intermediate evaluation, such as code generation with tests/compiler feedback. The main challenge outside math is obtaining a reliable evaluation signal.
>
> ## **W3: Related Work.**
> We agree that the distinction from BATS should be presented earlier. We will move this comparison to the Introduction / Method section and clarify that **BATS adapts prompt-level planning for tool-augmented agents**, whereas BG-MCTS makes the remaining budget an explicit input to tree-search decoding decisions (selection and widening).
>
> ## **W4: Sensitivity to Reward Model Quality**
> In this response, we additionally inspected the score distribution and found it to be highly saturated / near-binary (See the table below), which makes this a challenging weak-signal regime. Therefore, our experimental results in the submission suggest that BG-MCTS remains beneficial even with an imperfect PRM, and stronger PRMs would likely further improve both methods.
>
> | task | p10 | p25 | p50 | p75 | p90 |
> |---|---:|---:|---:|---:|---:|
> | Math500 lv.5 | $2.183 \cdot 10^{-3}$ | $9.740 \cdot 10^{-1}$ | $9.996 \cdot 10^{-1}$ | $9.988 \cdot 10^{-1}$ | $9.996 \cdot 10^{-1}$ |
> | AIME24/25 | $0.130 \cdot 10^{-4}$ | $1.170 \cdot 10^{-3}$ | $9.399 \cdot 10^{-1}$ | $9.968 \cdot 10^{-1}$ | $9.990 \cdot 10^{-1}$ |
>
> > *Table:* The percentiles of GenPRM evaluation scores observed when solving Math500 Level 5 and AIME24/25 under a fixed 30k-token budget using Qwen2.5-7B-Instruct.
>
> ## **W5/Q3/Q4: Fixed-budget formulations and extensions.**
> We adopt a fixed per-instance output budget in order to isolate the effect of budget-aware search within each problem. This does not preclude broader budget management. For dynamic allocation across instances, an outer controller can first assign a per-instance budget $B_i$, after which BG-MCTS runs unchanged with $\rho = 1 - C_{\mathrm{used}}/B_i$. Likewise, when long inputs consume a substantial part of the context window, the same mechanism applies by treating the remaining output/context allowance as the effective search budget, which simply causes the exploration-to-refinement transition to occur earlier. We will clarify both extensions in the revision.
>
> ## **Q1: Hyperparameter Sensitivity**
> In the paper, we keep $c=\sqrt{2}$ and $\kappa=\lambda=1$ fixed across all main experiments. In this response, we confirmed performance remains stable in a moderate neighborhood of the default setting (0.664--0.709 in our sweep on Qwen2.5-7B / Math500 at $B{=}20$K), and with a much smaller 5K budget BG-MCTS still exceeds standard MCTS (0.545 vs.\ 0.490 on Qwen2.5-7B / Math500 Lv.5). Because $\rho$ normalizes by the allocated budget, we expect better transfer across budget scales than a schedule tied to absolute token counts; domain transfer will depend more on the evaluation signal and step granularity than on retuning $\kappa,\lambda$.
> For a more detailed analysis of hyperparameter sensitivity, please see our response to Reviewer pWat.
>
> ## **Q5: Comparison with early stopping**
> LiteSearch's early stopping is already included in our main experiments; in Table~1, we only disable greedy pre-generation to keep token accounting consistent. Thus, the paper already compares against LiteSearch with early stopping, although we did not separately report applying the same stopping rule to BG-MCTS.
>
> To answer this point more directly, we additionally tested BG-MCTS with the same early-stopping rule on Llama-3.1-8B-Instruct / MATH500 (Lv.~5) / 20K. BG-MCTS+early stopping achieves 0.396 accuracy with 12.94K average tokens, compared with 0.291 / 16.40K for LiteSearch-Incre.\ and 0.271 / 16.16K for LiteSearch-Batch, while full-budget BG-MCTS achieves the highest accuracy overall (0.465). This suggests that early stopping and budget guidance are complementary.
>
> | Method | Accuracy (20K) | average cost |
> |---|---:|---:|
> | LiteSearch-Incre | .291 | 16.40k |
> | LiteSearch-Batch | .271 | 16.16k |
> | BG-MCTS(early stopping) | .396 | 12.94k |
> | BG-MCTS | .465 | 20.00k |
>
> > *Table:* Accuracy and average used token cost on MATH500 Level 5 using Llama-3.1-8B-Instruct under a fixed 20k-token budget. For BG-MCTS with early stopping, we use the same early-stopping criterion as LiteSearch.

---

> > ### Author Rebuttal · Reviewer_J8Nq · 2026-04-04
> >
> > The authors have provided satisfactory clarifications and additional evidence addressing all points raised in my original review, including the theoretical positioning, hyper-parameter sensitivity, and complementarity with early stopping.

---

> > > ### Author Response · Authors · 2026-04-06
> > >
> > > Thank you for your thoughtful follow-up and encouraging assessment. We are glad that our additional clarifications and evidence **fully addressed** your concerns regarding **the theoretical positioning, hyper-parameter sensitivity, and complementarity with early stopping.**

---

### Decision · Program_Chairs · 2026-04-30

**Decision:**

Accept (regular)

**Comment:**

This paper studies LLM tree-search decoding under fixed token budgets and proposes Budget-Guided MCTS, which uses the remaining budget to adapt both node selection and widening. Reviewers generally agreed that the setting is practically relevant and that the method is simple, well-motivated, and effective in the evaluated regime.

The main remaining concerns are that the empirical scope is still narrow (largely limited to math reasoning) and that the contribution should be positioned carefully relative to prior resource-aware search ideas. After rebuttal, however, these concerns appear to be more about scope and claim calibration than about core technical validity. The rebuttal added useful clarification on positioning, stronger-model evidence, and comparisons against simpler alternatives, which improves confidence that the gains are not merely due to a trivial scheduling heuristic.

Overall, while I agree that the paper should not overclaim generality or theoretical novelty, I find the contribution sufficiently clear and supported in the fixed-budget setting studied here. I therefore recommend acceptance.